# Satellite imagery reveals increasing volatility in human night-time activity

Tian Li[1,2 ✉], Zhuosen Wang[3,4], Christopher C. M. Kyba[5,6], Miguel O. Román[7], Karen C. Seto[8,9], Yun Yang[2], Shi Qiu[1], Theres Kuester[6], Michail Fragkias[10], Xiang Chen[11], Thomas H. Meyer[1], Chadwick D. Rittenhouse[1], Xiaonan Tai[12], Mari Cullerton[1], Falu Hong[1], Ashley Grinstead[1], Kexin Song[1], Ji Won Suh[1], Xiucheng Yang[1], Virginia L. Kalb[4], Chengbin Deng[13,14] & Zhe Zhu[1 ✉]

Artificial light at night (ALAN) marks the global impact of humanity[1,2]. Yet, our understanding of its true ebb and flow has been limited, often based on temporally aggregated satellite data that obscure finer dynamics. Here, using daily night-time satellite imagery[3] and a continuous change detection approach[4,5], we created global maps of high-frequency ALAN dynamics (2014–2022). Our findings challenge the prevailing perspective that changes in light radiance are largely gradual and unidirectional. Instead, the nightlights of Earth are surprisingly dynamic, characterized by frequent and coexisting brightening and dimming. On average, each location experiencing change underwent 6.6 distinct shifts over the 9 years. Driven by this volatility, the cumulative area of total ALAN change comprised 2.05 million km$^2$ of abrupt changes and 19.04 million km$^2$ of gradual changes. Brightening contributed a radiance increase equivalent to 34% of the 2014 global baseline, whereas dimming offset this by 18%. Notably, both brightening and dimming have markedly intensified over the past decade. This evidence of increasing volatility in human night-time activity provides an important dynamic dimension for understanding urban evolution, energy transitions, policy impacts and ecological consequences of rapidly changing illuminated nights.

The illuminated Earth, viewed from space at night, is a powerful testament to human presence, revealing a 'Black Marble' increasingly delineated by the light of human settlements, industries and energy infrastructures. Artificial light at night (ALAN) extends visibility beyond daylight hours, enabling round-the-clock movement, gathering and continuation of daily life. Yet, ALAN is far more than a visual spectacle: it is a direct, measurable signal of human activity, reflecting how we build and power our settlements, the dynamics of our economies and our responses to both crises and opportunities[6]. The variability of ALAN mirrors the pace and nature of human activity, manifesting as either abrupt events, such as new constructions or disasters, or gradual trends driven by long-term economic or demographic forces. Understanding the direction, location and intensity of these changes is, therefore, important for assessing the full scope of global change and its impact on human infrastructure and energy transitions[7]. We define brightening and dimming as sustained increases and decreases in radiance (excluding transient noise), respectively, driven by either abrupt events or gradual trends, as shown in Extended Data Fig. 5.

For decades, global assessment of ALAN trends has centred on a narrative of continuous and widespread brightening, with dimming viewed as a localized exception. This perspective stems largely from temporally aggregated night-time light (NTL) satellite observations[8,9]. Annual and multi-year composites, such as those produced from earlier Defense Meteorological Satellite Program Operational Linescan System (DMSP-OLS)[10] and later from Visible Infrared Imaging Radiometer Suite (VIIRS) Day/Night Band (DNB) data[11,12], have been instrumental in mapping long-term ALAN trends[2], documenting the expansion of urban extent[13] and estimating the light pollution effects[1]. Monthly VIIRS DNB products subsequently improved temporal granularity, enabling analyses of intra-annual variability[14], such as seasonal impacts[15] and epidemiological dynamics[16]. However, despite these strengths, temporal aggregation inherently masks short-term fluctuations and the bidirectional changes arising from the coexistence of brightening and dimming.

In reality, human activities and the resulting changes in ALAN are not uniform, linear or steady processes. They operate across a spectrum of timescales, ranging from gradual shifts such as suburban expansions or LED adoptions to discrete, abrupt events such as industrial constructions, changes in municipal lighting codes, power outages triggered by disasters and the disruptions of infrastructure or destructions of

[1]Department of Natural Resources and the Environment, University of Connecticut, Storrs, CT, USA. [2]School of Integrative Plant Science, Cornell University, Ithaca, NY, USA. [3]Earth System Science Interdisciplinary Center, University of Maryland College Park, College Park, MD, USA. [4]Terrestrial Information Systems Laboratory, NASA Goddard Space Flight Center, Greenbelt, MD, USA. [5]Institute of Geography, Ruhr University Bochum, Bochum, Germany. [6]GFZ Helmholtz Centre for Geosciences, Telegrafenberg, Potsdam, Germany. [7]Earth Sciences Division, NASA Goddard Space Flight Center, Greenbelt, MD, USA. [8]Yale School of the Environment, Yale University, New Haven, CT, USA. [9]Hixon Center for Urban Sustainability, Yale University, New Haven, CT, USA. [10]Department of Economics, College of Business and Economics, Boise State University, Boise, ID, USA. [11]Department of Geography, Sustainability, Community and Urban Studies, University of Connecticut, Storrs, CT, USA. [12]Department of Biological Sciences, New Jersey Institute of Technology, Newark, NJ, USA. [13]Center for Spatial Analysis, University of Oklahoma, Norman, OK, USA. [14]Department of Geography and Environmental Sustainability, University of Oklahoma, Norman, OK, USA. ✉e-mail: tianli@uconn.edu; zhe@uconn.edu

buildings due to armed conflicts[17-20]. Focusing only on the net change derived from temporal composites masks the specific timing and impacts of these incidents. This oversight limits our ability to truly link cause and effect, assess policy effectiveness with precision, and understand the full ecological implications of a rapidly changing nocturnal environment.

More recently, advances in NASA's daily Atmospheric- and Lunar-BRDF (bidirectional reflectance distribution function)-corrected Black Marble NTL product[3] have provided an unprecedented opportunity to quantitatively characterize the finer daily night-time light dynamics of Earth. Unlike earlier nightlight products that primarily served visualization or coarse trend analysis, Black Marble applies comprehensive corrections for atmospheric conditions, terrain and lunar illumination to improve radiometric stability, and at the same time provides information on the contaminated pixels, such as clouds and snow[21]. This enables consistent detection of subtle yet meaningful changes that reflect real-world dynamics, making it a trusted source of information for stakeholders. However, this abundance of daily data comes with its analytical hurdles. The signal is subject to substantial noise from a variety of sources, including atmospheric interference, variations in sensor viewing and local geometry, and ephemeral conditions such as snow cover[22]. The magnitude of this combined noise is highly variable and can often exceed the subtle, real-world changes we aim to detect, making most current algorithms ineffective at separating the true signal from the noise.

This study presents the first comprehensive global analysis of ALAN change dynamics derived from daily Black Marble NTL data. By adapting a continuous change detection algorithm[4,5] (Methods), we quantified the timing (day-of-year and year), intensity (area-averaged radiance change), type (abrupt or gradual), and direction (brightening or dimming) of ALAN changes for every 15-arc-second pixel (about 500 m at the equator) across the primary inhabited landmasses of Earth (70° N–60° S) from 2014 to 2022. By tracking each distinct change event, our analysis captures the full trajectory of development and decline over time, explicitly accounting for pixels experiencing multiple changes.

We analysed 1.16 million daily NASA Black Marble NTL images constrained to the analysed NTL areas (15.16 million km², 10% of global land; Supplementary Fig. 3 and Supplementary Information Section 1), excluding persistently dark regions and areas with ephemeral natural light events that did not meet our persistence criteria (Methods). Validation against independent stratified random sample units ($n = 2,071$ for abrupt changes and $n = 1,902$ for gradual changes) confirmed global reliability (Fig. 1 and Extended Data Table 1). To quantify these dynamics, we analysed the area of change, the radiance change (overall shift in light output) and the change intensity. Unless specified, all regional and global values were derived from the native 15-arc-second grid results projected to a '500-m' nominal resolution equal-area sinusoidal grid.

## Dynamics and drivers of ALAN change

Our data reveal a dynamic night-time environment in which changes are frequent rather than sporadic. Over 2014–2022, the ever-changed area, defined as the total area experiencing at least one ALAN change, was substantial (3.51 million km²; Supplementary Table 1). Globally, although nearly half (51%) of these altered areas saw only gradual changes, more than one-third (35%) experienced both abrupt change events and longer-term gradual changes, with the remainder (14%) marked solely by abrupt changes (Extended Data Table 2). Further analysis at the country or territory level reveals diverse national profiles in the typical frequency of abrupt and gradual changes (Supplementary Fig. 1).

More revealing of the true dynamics is the total cumulative ALAN change area (or gross area change) during 2014–2022 (Extended Data Table 3), which sums the area changed each year, thereby accounting

for pixels that underwent several changes (for example, a brightening followed by a dimming, or multiple abrupt events). Unbiased area estimates indicate that between 2014 and 2022, 2.05 (95% confidence interval (CI) [1.79, 2.32]) million km² underwent abrupt ALAN changes, and 19.04 (95% CI [18.10, 19.98]) million km² experienced gradual changes. Asia, particularly China and India, accounted for the largest cumulative area of ALAN change. This cumulative change area over the 9 years is 5.5 times the global lit area of the 2014 baseline, with an average of 6.6 changes per ever-changed area, indicating that each parcel of lit land experienced several significant ALAN alterations during the study period (Fig. 2a,b, pie charts).

The high frequency of abrupt change underscores the transient nature of ALAN signals across the globe, with more than 20% of affected areas experiencing abrupt changes more than once (Fig. 2a). The spatial pattern, observed in many parts of the world, reflects ongoing cycles of construction and demolition (for example, in China[23] and India[24]), energy instability (for example, grid failures in Venezuela[25] and energy crises in Lebanon[26]), fossil fuel operations (for example, gas flaring in Texas, USA[27]) or societal disruptions (for example, conflicts in the Middle East[17]) (Fig. 2a). By contrast, gradual ALAN changes reveal long-term trends, with 94% of affected areas experiencing changes that persisted for more than 1 year, particularly developed regions such as the USA and Europe (Fig. 2b). These gradual shifts are often tied to slow demographic-economic dynamics, dedicated dark-sky conservation efforts (for example, the UNESCO Starlight Reserves), or systematic infrastructure upgrades (for example, LED replacement programs of Europe[28]). The only places in which lighting remains mostly unchanged are uninhabited regions and regions with limited development, such as protected natural reserves and remote deserts (Fig. 2b).

Importantly, those changes are shifting in both directions, with brightening accounting for 65% of abrupt and 71% of gradual changes, whereas dimming accounts for 35% of abrupt and 29% of gradual changes (Fig. 2c). Using the Stokes and Seto framework[29] (Extended Data Table 4), we found more than half of the abrupt brightening was driven by non-residential development and electrification, highlighting the role of infrastructure expansion and rural electrification in shaping short-term ALAN spikes. By contrast, abrupt dimming was mainly attributed to reductions in gas flaring (46%), driven by government regulations, gas infrastructure upgrades and operational volatility[30]. Gradual changes followed a similar brightening–dimming ratio but with distinct leading drivers. Gradual brightening was often tied to concurrent change, such as steady urban expansion, whereas gradual dimming was primarily caused by de-electrification, reflecting long-term declines in lighting infrastructure or energy access. Moreover, the global bidirectional patterns indicate that frequent brightening regions were primarily driven by concurrent change and non-residential development, typically reflecting rapid urbanization and development (for example, Eastern and South Asia[31]), or emerging economies expanding energy access (for example, West Africa[32]) (Extended Data Fig. 1a). Conversely, regions with frequent dimming reflect energy conservation initiatives in high-income countries/territories (for example, Eastern USA, Western Europe[33]) or systemic instability, such as the rolling blackouts caused by load shedding in South Africa[34] (Extended Data Fig. 1b).

## Heterogeneous geography of ALAN changes

The long-term global trajectory points towards a brighter planet, evidenced by a net 16% increase in total ALAN radiance from 2014 to 2022 (Fig. 3e), outpacing global population growth[35]. Specifically, brightening contributed a radiance increase equivalent to 34% of the 2014 baseline, while dimming offset this by 18%. Yet, this aggregate figure conceals a widespread coexistence of brightening and dimming (Fig. 3). Mapping the spatial distribution of ALAN change area and change intensity between 2014 and 2022 reveals substantial regional heterogeneity

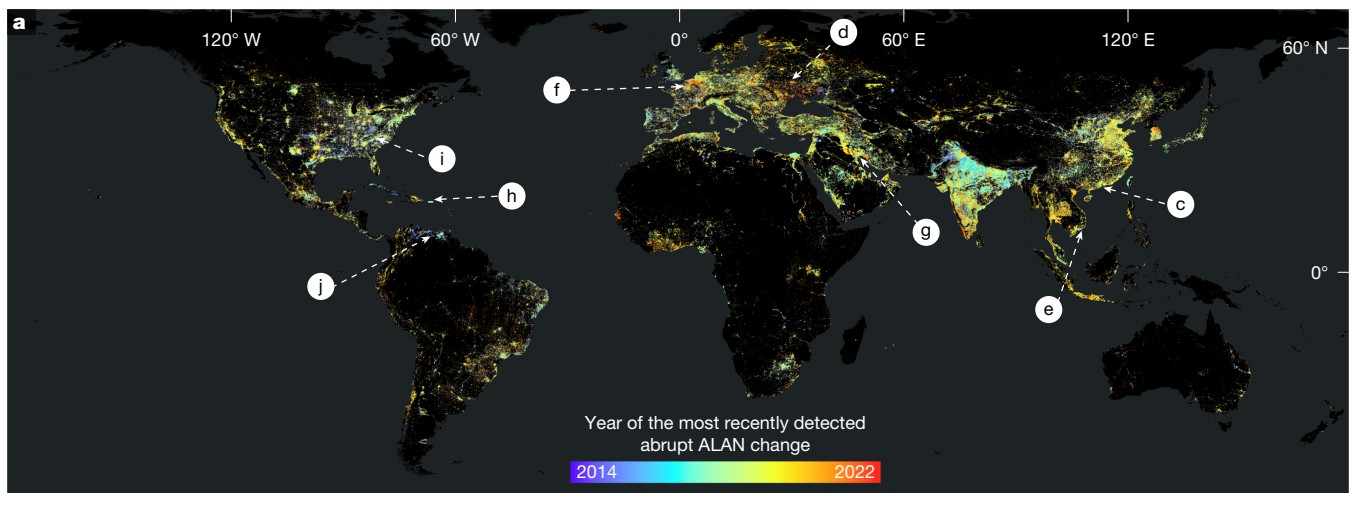

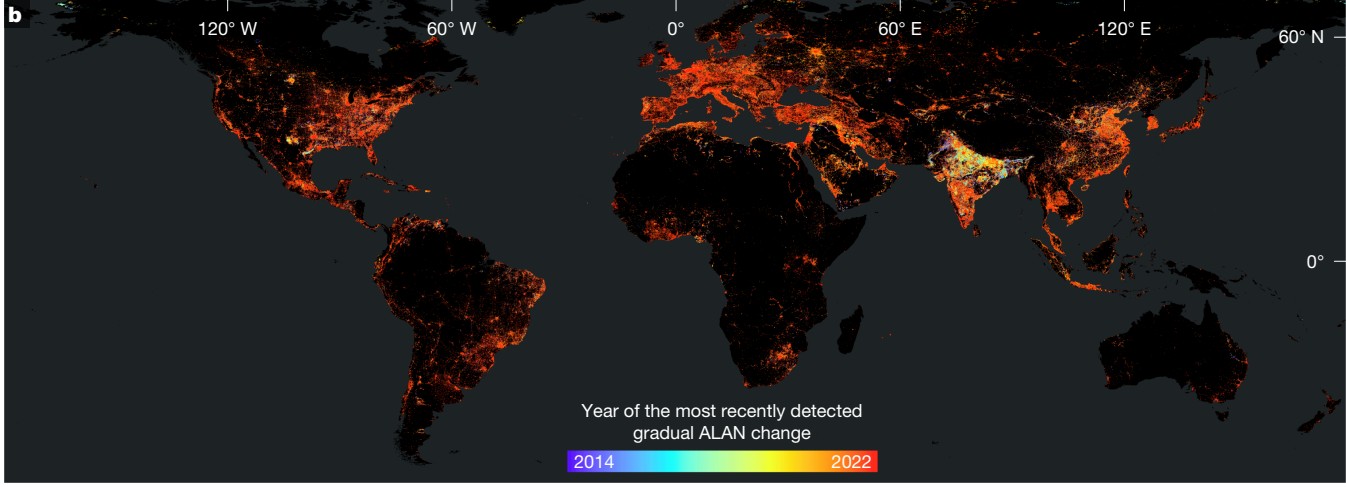

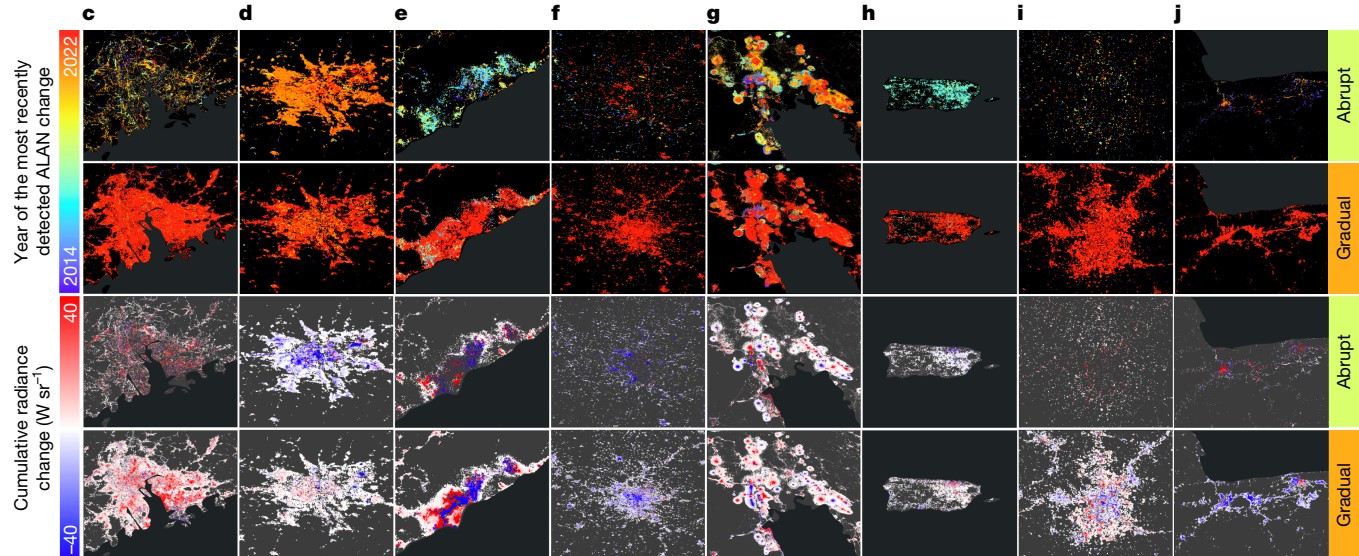

**Fig. 1 | Global ALAN change product (2014–2022). a**, The accumulated ALAN change time map for abrupt changes. The colours show the year of the most recently detected abrupt changes. **b**, The accumulated change time map for gradual changes. The colours show the year of the most recently detected gradual changes. **c–j**, Magnified examples showing the accumulated most recent abrupt and gradual ALAN change time and the cumulative radiance change caused by different driver types over the 9 years (locations indicated by arrows in **a**). Urbanization processes in Guangzhou, China (**c**); armed conflicts in Kyiv, Ukraine (**d**); development of dragon fruit agriculture in Vietnam (**e**); environmental policies in Paris, France (**f**); gas flare changes in the Middle East region (**g**); power outages caused by hurricanes in Puerto Rico (**h**); urban expansion and decentralization in Charlotte, USA (**i**); and dimming caused by economic collapse in Valencia and Caracas, Venezuela (**j**). The maps in **a** and **b** are shown in 0.05° × 0.05° grid cells, with mean aggregation for visualization, whereas the maps in **c–j** are in the original 15-arc-second resolution. The global ALAN change product can be accessed by GitHub (https://github.com/GERSL/VZA-COLD). Basemaps in all panels are adapted from the World Continents Layer from Esri.

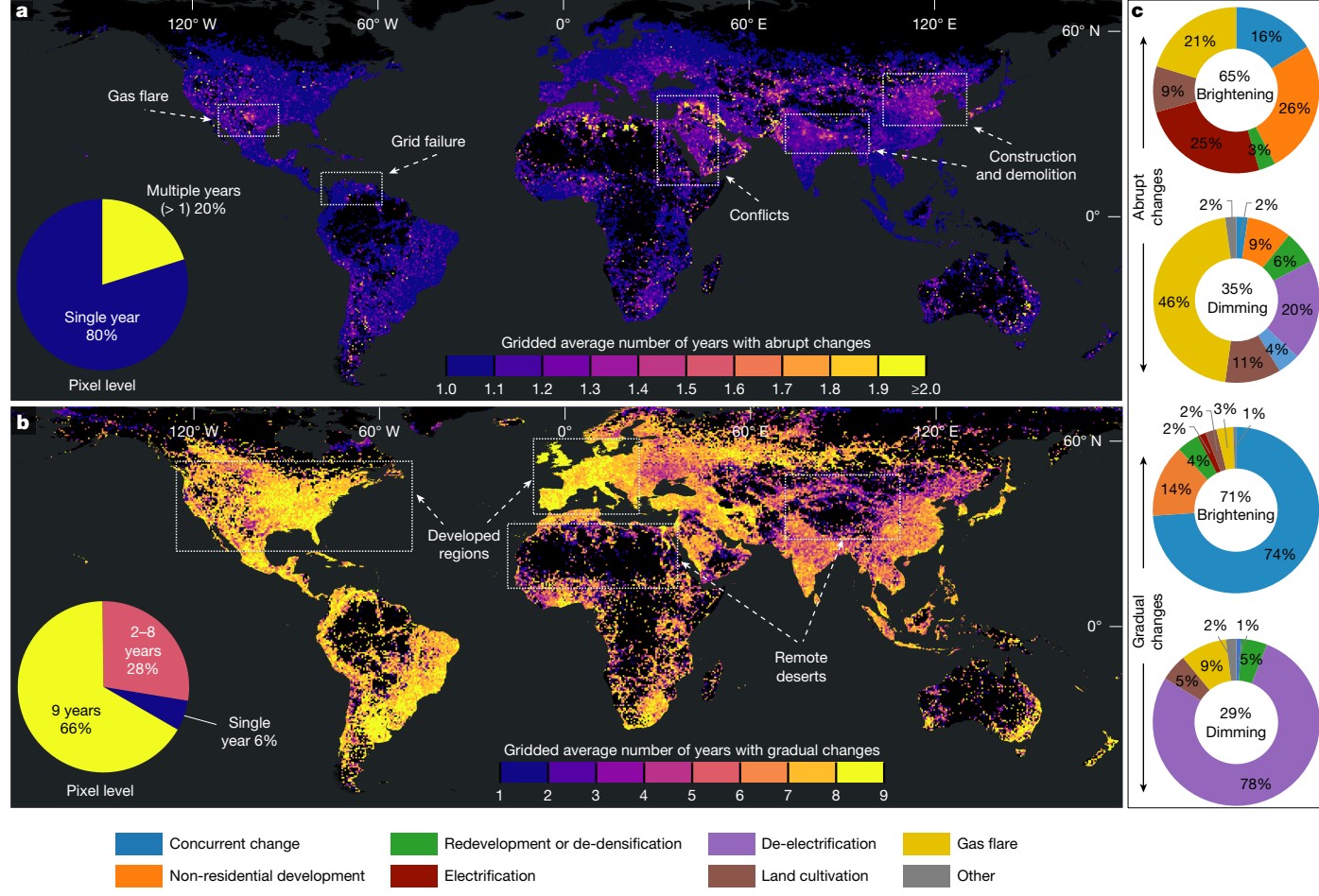

**Fig. 2 | Temporal frequency and estimated causal drivers of global ALAN changes from 2014 to 2022. a,b**, Maps show the average number of years (per 15-arc-second pixels within 0.5° × 0.5° grid cells) experiencing abrupt ALAN change (**a**) and gradual ALAN change (**b**). **c**, Donut charts showing the estimated proportions of causal drivers associated with the brightening and dimming for both abrupt and gradual changes. The central percentage in each donut chart indicates the proportion of brightening compared with dimming cumulative change areas derived from the abrupt and gradual change maps (Supplementary Table 2), whereas the segment values on the donut charts represent driver proportions estimated from our interpretation of the validation sample (Extended Data Table 1). Basemaps in **a** and **b** are adapted from the World Continents Layer from Esri.

(Extended Data Fig. 2 and Supplementary Fig. 2). Broad patterns of brightening are evident across the world (Supplementary Table 3), led by Asia, and reflecting continued urbanization, industrial expansion and rural electrification. Although China and India show the largest national-level increases (Fig. 3a, L1), our data show strong regional contrasts. In China, brightening is concentrated in the eastern and central regions, driven by urbanization and industrial activity, whereas western areas show lower levels of change and more spatially fragmented patterns. In India, southern regions experienced sustained brightening throughout the study period, reflecting higher levels of urbanization and economic development, whereas northern regions exhibited brightening primarily in the early years (Fig. 1a,b), driven by national rural electrification and street lighting programmes that expanded power access and lighting infrastructure. Much of Sub-Saharan Africa also shows a strong brightening signal, reflecting development that illuminates previously unlit regions[36].

Substantial dimming patterns are also observed in several regions (Fig. 3a). Europe presents a particularly clear and structured dimming pattern, with a 4% net decrease in ALAN radiance relative to its 2014 baseline. Notably, dimming alone accounts for an 18% reduction, and these changes align closely with national borders, highlighting the impact of country-specific lighting regulations. Extensive areas experienced these reductions, most notable in France (33% net drop), the United Kingdom (22%), and the Netherlands (21%) (Fig. 3a, L2, and

Supplementary Fig. 2). This reflects a combination of widespread technological shifts from older, less efficient lighting to newer LED systems, measures to reduce light pollution and energy use, and broader national and EU-level energy efficiency mandates[18,37,38].

By contrast, the dimming observed in Venezuela (Fig. 3a, L3) is not driven by regulation or technology but stems from systemic collapse. Here, ALAN radiance declined by more than 26% relative to its 2014 baseline, reflecting economic downturns, widespread infrastructure decay and lack of investment[39]. These cases highlight how reductions in night-time illumination can arise not only from deliberate energy-saving or pollution-mitigation strategies but also from infrastructural breakdown and economic instability.

The USA offers a microcosm of this complexity (Fig. 3a, L4). The West Coast brightened with ongoing population growth and vibrant economies in its main urban centres[40], whereas the East Coast and parts of the Midwest dimmed because of de-densification in some older urban cores[41], the decline of certain manufacturing sectors and the adoption of energy-efficient lighting technologies. Central USA regions, particularly areas overlying the main oil and gas shale basins such as the Permian (Texas) and Bakken (North Dakota), exhibit highly volatile ALAN signals. These regions exhibit strong bidirectional changes: intense brightening during boom periods due to drilling activity and gas flaring, followed by equally sharp dimming as operations scale down or shift location[27]. This reflects oil and gas extraction cycles,

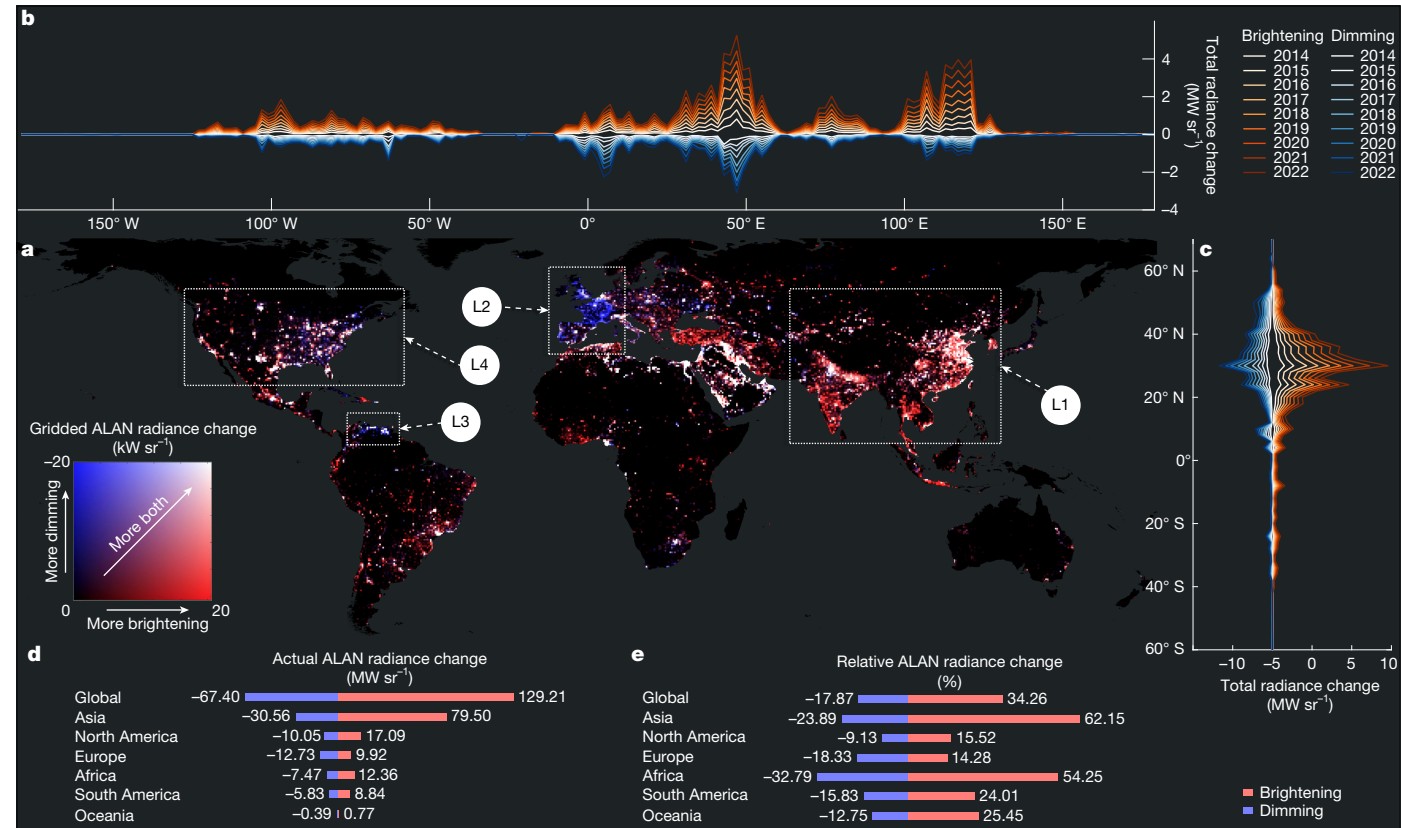

**Fig. 3 | Global patterns of ALAN radiance change from 2014 to 2022.**
**a**, Gridded heat map showing the sum of ALAN radiance changes. Values represent the sum of all brightening radiance changes (from black to red) and the sum of all dimming radiance changes (from black to blue) within each 0.5° × 0.5° grid cell. Brighter colours mean larger change in both directions, and darker colours mean the opposite. Labelled regions (L1–L4) highlight key regional patterns of brightening, dimming and bidirectional change discussed in the text. **b**,**c**, Longitudinal (**b**) and latitudinal (**c**) cumulative profile plots of the yearly sum of brightening (red/orange tones) and yearly sum of dimming (blue/cyan tones) ALAN radiance changes in every 2° step. **d**, Absolute ALAN radiance change summed at global and continental levels for brightening and dimming, respectively. **e**, Relative ALAN radiance changes by the end of 2022, compared with the baseline radiance on 1 January 2014, at global and continental levels for brightening and dimming, respectively. The red bars in **d** and **e** represent the brightening changes, and the blue bars represent the dimming changes. Basemap in **a** is adapted from the World Continents Layer from Esri.

in which the timescale of lighting changes is often dictated by the specific methods and phasing of extraction activities, rather than broader oil price fluctuations alone. Similar dynamic signals are observed in oil-producing regions across the Middle East.

## Change intensity compared with change area

The ALAN change intensity quantifies the average strength of those changes, whereas change area captures how broadly illumination shifts spread (Extended Data Fig. 2). This distinction differentiates widespread but subtle changes from more localized but intense transformations. Globally, a complex picture emerges when examining these intensities relative to the 2014 baseline (Supplementary Table 4). Areas experiencing brightening saw their average intensity increase by 9%, whereas those experiencing dimming saw a decrease of 10%.

Regionally (Extended Data Fig. 2f–j), Asia and Africa exhibit the most intense bidirectional change intensities (brightening: 10–11% relative to their initial radiances; dimming: −11% to −15%), underscoring them as hotspots of strong development and retreat. By contrast, the USA showed widespread but moderate change intensities (brightening: 8% of initial radiance; dimming: −6%). Apart from high-intensity oil extraction regions, shifts elsewhere reflect incremental, low-density suburban sprawl[42]. Unlike high-density redevelopment, this horizontal expansion is additive, distributing artificial light across vast peri-urban areas without requiring the disruption of existing infrastructure, resulting

in widespread but gradual trends[43]. India presents a similar profile of widespread cumulative change area accompanied by moderate change intensities. This contrasts sharply with China, where high-intensity signals are consistent with a strategy of vertical urbanization and high-density land conversion[44]. This specific evolution of land use creates a unique NTL signature with periods of dimming (demolition) followed by explosive brightening (vertical reconstruction). Consequently, higher urban density acts as a multiplier for NTL volatility, whereas lower-density sprawl acts as a dampener.

## Intensifying bidirectional dynamics

Temporal trends in radiance changes (2014–2022) show a critical evolution of global ALAN dynamics (Fig. 4). Although the global total net radiance change trended upward ($P < 0.05$), this conceals an intensification ($P < 0.05$) of both brightening and dimming over years (Fig. 4a), indicating that the global nightscape is becoming more dynamic and volatile overall. Countries and territories undergoing net brightening are often not doing so in a uniform, one-sided way (Supplementary Table 5). In most cases, both brightening and dimming radiance changes are increasing simultaneously (Fig. 4b). This bidirectional dynamism is pronounced in East Asia and Africa, where expanding light footprints coexist with intensifying pockets of dimming. By contrast, the USA and Australia show significant ($P < 0.05$) intensification only in brightening. Meanwhile, most of Europe, South Africa and New Zealand exhibit intensifying bidirectional trends despite their net dimming trajectories.

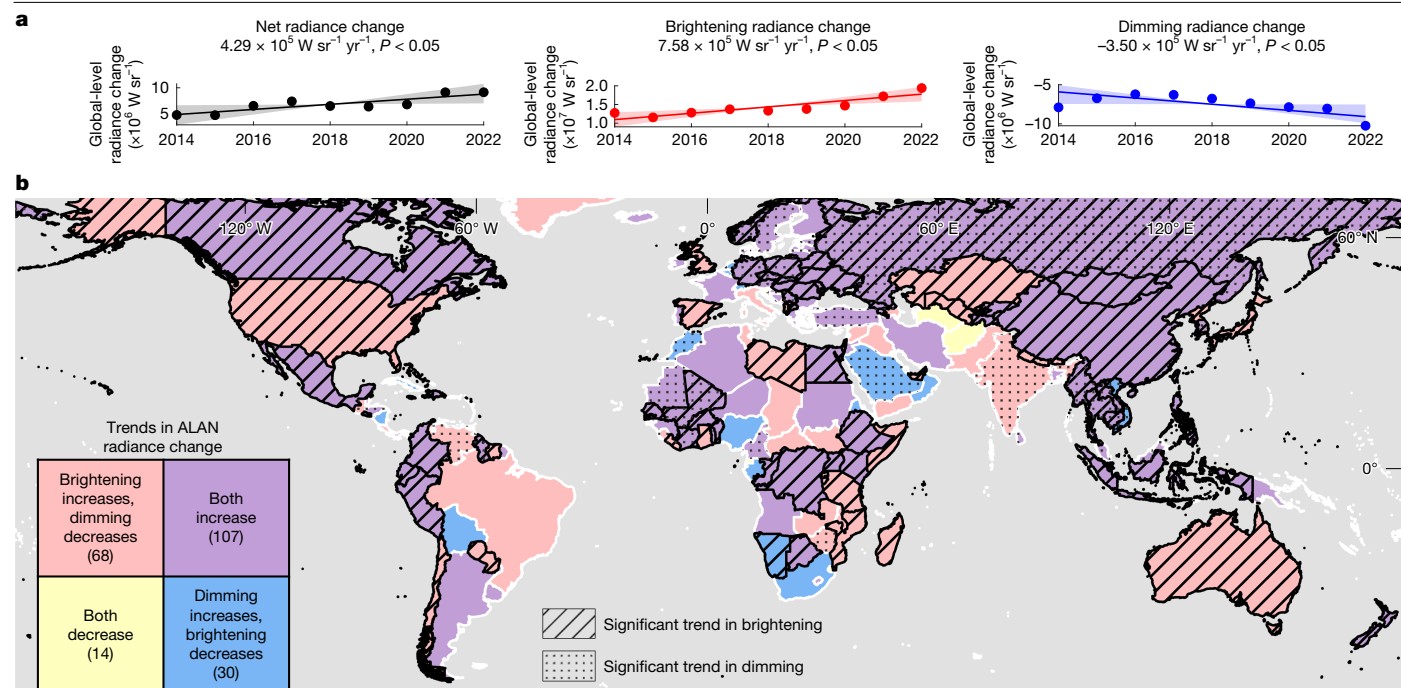

**Fig. 4 | Temporal trends of global and country- and territory-level ALAN radiance change from 2014 to 2022. a**, Line plots showing the global trends in annual ALAN radiance change: total net radiance change (left); total brightening radiance change (middle); and total dimming radiance change (right). The dots show the corresponding global annual sum of ALAN radiance changes from 2014 to 2022, the solid lines show the overall Sen's slope trends and the shaded areas represent their 95% CI. **b**, Bivariate map showing the trends in annual radiance change of brightening (increasing compared with decreasing) and trends in annual radiance change of dimming (increasing compared with decreasing). Countries and territories with a trend equal to zero are not shown. Map colours represent the directions of trends in annual radiance change of brightening and dimming, the stripe texture indicates regions with statistically significant trends in brightening changes and the dot texture indicates regions with statistically significant trends in dimming changes (Sen-slope, Mann–Kendall $P < 0.05$). Numbers in the legend show the total number of countries and territories with different trends in brightening and dimming radiance changes. Basemap in **b** is adapted from the World Continents Layer from Esri.

This intensification of bidirectional change is further corroborated by disaggregated global trends in change area and change intensity. The global annual area experiencing dimming grew significantly (expanding by 12,875 km² yr⁻¹, $P < 0.05$) (Extended Data Fig. 3a, right), driven by gradual shifts in Africa, Asia, Europe and South America (Extended Data Fig. 3b and Extended Data Fig. 4c). Conversely, the brightening area showed a slight, non-significant decrease (Extended Data Fig. 3a, middle). In terms of change intensity, the global brightening trend increased significantly (increasing by 0.04 nW cm⁻² sr⁻¹ yr⁻¹, $P < 0.05$) (Extended Data Fig. 3c, middle). Meanwhile, the intensity of dimming events, led by abrupt events (Extended Data Fig. 4b), also showed a negative trend of −0.03 nW cm⁻² sr⁻¹ yr⁻¹ (Extended Data Fig. 3c, right).

## Capturing dynamics of night by daily data

Daily NTL observations offer an unprecedented window into societal dynamics, capturing short-term events often lost in coarser temporal composites (Fig. 5). This high-frequency perspective allows us to monitor the dynamics of night—the rapid and often abrupt fluctuations of human activity and societal responses as reflected in ALAN. Although annual and monthly data track broad long-term trends in energy use or ecological impact[8], our daily-derived ALAN changes show crucial short-term, event-driven dynamics that shape those net long-term outcomes. By resolving the precise timing of these changes, we can move beyond broad correlations to link shifts in illumination directly to specific real-world shocks (for example, military actions, lockdowns or policy implementations), while capturing the full magnitude of transient events (for example, Fig. 1h) that would otherwise be smoothed out in monthly averages.

This granular analysis exposes societal dynamics and volatility that are often masked by net trends. A central insight from this study is that regions with comparable annual outcomes can behave very differently at finer timescales. For instance, areas with high fluctuation often signal instability (for example, armed conflicts in Syria), reflecting energy insecurity, economic precarity or sporadic conflict, even if their net trend appears stable. Conversely, low fluctuation suggests steady, uninterrupted development or intentional dimming (for example, Western Europe). Acting as a societal electrocardiogram, these dynamics are important for distinguishing resilient systems and those under acute stress.

Continent-level analysis shows a notable increase in ALAN volatility after 2020, evident in both brightening and dimming events, with dimming showing the steepest downturn (Fig. 5a). This growing volatility reflects a convergence of main global and regional disruptions, including COVID-19 lockdowns, accelerated LED transitions, light pollution policies and the energy crisis triggered by the Russia–Ukraine war. A prominent example is the global dip in ALAN radiance, primarily driven by the abrupt dimming events in Asia, which experienced the earliest and most extensive lockdowns during COVID-19 (ref. 45) (Fig. 5b). This sudden decrease, visible with varying degrees across multiple continents in early 2020, aligns precisely with widespread lockdowns and curtailments of economic and social activity during the initial wave of the pandemic (Fig. 5d). Another compelling signal emerges in Europe during 2022 (Fig. 5c), in which a sharp and sustained decrease in ALAN radiance change is observed, contrasting with trends in most other continents during the same period. The timing aligns with the Russia–Ukraine conflict and subsequent European energy crisis, leading to widespread energy-saving measures across many European nations, notably France and Belgium[46] (Fig. 5e).

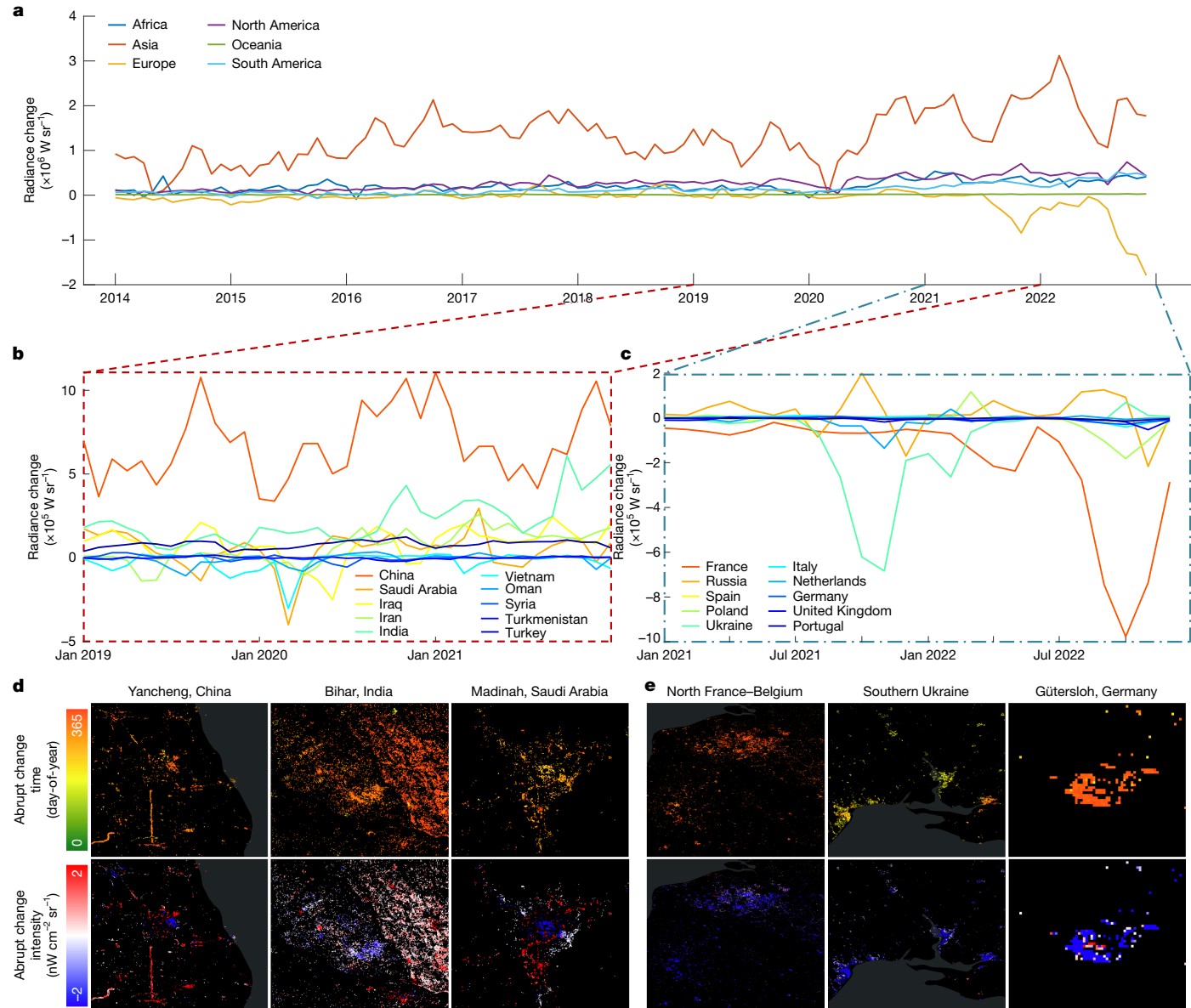

**Fig. 5 | Daily NTL-based ALAN changes show the dynamics of night by highlighting specific emergent changes. a**, Continental-level monthly net radiance changes from 2014 to 2022. **b**, Country- and territory-level monthly net radiance changes from 2019 to 2021 for the top 10 Asian countries and territories exhibiting the most notable dimming in 2020. **c**, Country- and territory-level monthly net radiance changes from 2021 to 2022 for the top 10 European countries and territories with the most intensive dimming in 2022. **d**, Spatial examples: day-of-year abrupt ALAN change timing and change intensity maps for regions substantially affected by COVID-19 lockdowns in 2020. Examples include notable dimming in the manufacturing zone of Yancheng, China;

industrial areas in Saran district, Bihar, India; and national-level lockdowns in Madinah, Saudi Arabia. **e**, Spatial examples: day-of-year abrupt ALAN change timing and change intensity maps of regions affected by the European energy crisis and armed conflicts in 2022. Examples include policy-driven light adjustments in northern France; dimming in energy-intensive industrial areas in Flanders, Belgium; distinct dimming timings corresponding to the Russia–Ukraine conflict progression in Mykolaiv, Odesa, and Kherson, Ukraine; and streetlight reductions in Gütersloh, Germany. Basemaps in **d** and **e** are adapted from the World Continents Layer from Esri.

## A new view of our illuminated planet

This global, high-resolution analysis of ALAN dynamics, derived from daily NASA Black Marble satellite observations, refines and expands our understanding of how humanity is altering the night environment[3]. Our findings show that the human light footprint is not a universally expanding entity but a dynamic system, characterized by the pervasive coexistence of brightening and dimming. The concurrent rising trends in the change area of dimming, the radiance change and the change intensity point to an intensification of this bidirectional dynamic. This signals an acceleration of the processes that modify the night-time environment, driven by increasingly potent

and often counteracting forces, rendering the nocturnal world more volatile.

Consequently, our findings demand a re-evaluation of ALAN as a socioeconomic proxy. The inherent complexity and bidirectional nature of ALAN change are especially evident in regions undergoing rapid technological transitions, strong policy interventions or economic instability. Thus, simple correlations between net ALAN radiance change and indicators such as the gross domestic product may be misleading. These events act as exogenous shocks to the socio-technical system, creating dynamic, time-lagged interdependencies that require robust empirical modeling[47]. Methodologies such as the simultaneous analysis of multivariate time series are essential here. Specifically, advanced

econometric tools analysing impulse response functions offer the necessary framework to disentangle these complex interactions[48].

Moreover, our daily-derived, disaggregated dataset enables more sophisticated and accurate modelling. Understanding socioeconomic responses to ALAN reductions or fluctuations, not just increases, is an important, largely unexplored frontier, which requires such spatially and temporally explicit information. Furthermore, the demonstrated sensitivity to events such as pandemics and conflicts highlights the potential for near-real-time monitoring of societal disruptions, disaster impacts and infrastructure resilience. This ability, potentially extending to early warnings for economic cycles[49], positions daily ALAN analysis as a vital tool for informing policy, humanitarian aid and ecological response[50], ultimately strengthening societal resilience and guide resource allocation.

Finally, it is crucial to recognize that current operational satellite NTL observations, such as those from VIIRS DNB, primarily capture post-midnight light emitted upwards from the surface, whereas ground-based monitors similar to those used for the World Atlas measure downward skyglow[1]. These represent distinct physical quantities that are not directly comparable but provide complementary information regarding the night environment. Moreover, VIIRS DNB is most sensitive to a specific spectral range (around 500–900 nm), leaving a blind spot for trends occurring earlier in the evening or in different spectral bands[1,6]. Yet, beyond these observational nuances, the overarching signal is unmistakable: the Black Marble of Earth is not merely growing brighter; it is pulsing with intensifying volatility, echoing the amplifying heartbeat of human activity.

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

## Methods

### Data and study area

The foundational dataset for this global analysis of ALAN dynamics was the Black Marble product suite of NASA (Collection 1), derived from the DNB sensor onboard the VIIRS instrument[3]. The DNB observes light in the wavelength range of 500–900 nm, with an equatorial local overpass time of roughly 1:30 a.m. Standardized quality assurance of the Black Marble data ensures consistency across time, geography and sensors (Suomi-NPP and the NOAA-20/NOAA-21VIIRS DNB), enabling cross-mission compatibility vital for continuous monitoring applications[51]. We acquired two specific daily products (that is, VNP46A2 and VNP46A1) from the NASA Level-1 and Atmosphere Archive and Distribution System Distributed Active Archive Center (LAADS DAAC), covering the period from 1 January 2013 to 31 December 2023.

The primary daily NTL data for detecting changes was the VNP46A2 product, which provides daily ALAN radiance values (nW cm$^{-2}$ sr$^{-1}$) corrected for atmospheric influences and bidirectional reflectance distribution function (BRDF) effects from lunar illumination geometry and diverse surface reflectance variability[3]. This rigorous design improves radiometric stability and distinguishes Black Marble from earlier NTL products, supporting quantitative, science-quality analysis. Data are provided in geographic coordinates (WGS84) at a nominal spatial resolution of 15 arc-seconds (about 460 m at the equator), finer than the intrinsic sensor resolution of about 750 m. Complementing this, we used the VNP46A1 products to provide the pixel-level quality assessment flags of cloud, snow/ice and solar/lunar contamination information[21]. These products also supplied detailed viewing geometry, specifically the sensor viewing zenith angle (VZA), a key parameter for the VZA-stratified COntinuous monitoring of Land Disturbance (VZA-COLD) change detection algorithm used to mitigate angular effects on observed NTL[4].

To ensure high-quality inputs for the change detection algorithm and improve processing efficiency, two pre-processing steps were conducted: systematic filtering of low-quality daily images and masking of persistently dark areas with no historical artificial light (Supplementary Information Section 1). The ALAN change detection results were generated under the linear latitude/longitude geographic projection, consistent with the input atmospheric- and lunar-BRDF-corrected Black Marble data. This projection was also retained for visualizations of the global maps. However, all quantitative analyses in this study, such as area estimates and accuracy assessments, were conducted following the MODIS/VIIRS 500-m nominal resolution sinusoidal equal-area projection to ensure accurate area-based calculations.

The geographic scope of this study was defined as the global terrestrial regions located between 70° N and 60° S. This latitudinal range was chosen because it encompasses the vast majority of the landmasses of Earth and virtually all primary human settlements and areas of substantial ALAN[52]. Polar regions were excluded from the analysis because of data acquisition challenges (for example, polar day, extensive snow and ice). Oceans and large inland water bodies were masked using the Terra Moderate Resolution Imaging Spectroradiometer (MODIS) Land Water Mask product in 2014 (MOD44W Collection 6.1)[53]. The World Bank Official Boundary data (accessed 1 January 2025) were used for regional aggregation (continents, countries, and territories). The core period for ALAN change detection was 1 January 2014 to 31 December 2022. Data from 2013 served for model initialization, and data from 2023 for confirming end-of-series changes (see details in Supplementary Information Section 1).

### Definition of ALAN change types

In this study, ALAN changes are categorized into two primary types based on their temporal patterns: abrupt and gradual (Supplementary Fig. 4c). These are further classified as either brightening or dimming, depending on the direction of change in radiance. Abrupt ALAN changes are short-term shifts characterized by sudden step-like changes in NTL radiance or a structural break in the time series. These changes typically unfold over a span of weeks to months and often correspond to discrete events such as urban construction or demolition (Fig. 1c), natural disasters (Fig. 1h), or armed conflicts (Fig. 1d). Abrupt changes also include changes that caused sudden redirections of a longer-term trend, such as the onset of economic recession or a surge in foreign investment or immediate policy actions (for example, beginning of rapid lighting installation), leading to a noticeable inflection in the trajectory of ALAN dynamics (Fig. 1j). By contrast, gradual ALAN changes represent long-term, continuous trends that unfold over periods exceeding 1 year. These changes exhibit a relatively stable, directional pattern, either brightening or dimming, without abrupt discontinuities. Gradual changes typically reflect sustained socioeconomic or demographic processes such as rural expansion (Fig. 1i), economic transformation or the systematic rollout of new lighting technologies (for example, LED retrofitting). Unlike abrupt events, they result in a smooth and persistent evolution in night-time radiance over time.

Each ALAN change is further classified by its direction (that is, brightening and dimming) based on Extended Data Table 5 (equations (4) and (6)). Brightening corresponds to a positive model-estimated change in radiance, whereas dimming corresponds to a negative change. This directional classification is essential for disentangling the complex global patterns of illumination gain and loss, enabling a more complete understanding of the dynamic behaviour of ALAN across space and time.

Although these persistent changes are the focus of our analysis, transient fluctuations that dissipate within days to a few weeks, such as those caused by temporary outages, meteorological anomalies or daily variations in power supply, are not analysed. These ephemeral changes, which typically return to the original status of the NTL intensity within 1 month, fall outside the analytical scope of this study. Our emphasis is on sustained alterations in ALAN radiance rather than day-to-day variability in the signal. The workflow of our method and analysis is shown in Extended Data Fig. 5.

### ALAN change metrics calculation

The core methodology for detecting changes in ALAN was based on the VZA-COLD algorithm, detailed in ref. 4 and Supplementary Information Section 2. VZA-COLD mitigates viewing geometry effects by stratifying daily NTL observations for each pixel into four VZA intervals (0°–20°, 20°–40°, 40°–60°, 0°–60°). For each stratum, a harmonic time series model is continuously fitted using robust regression to capture intra-annual seasonality and inter-annual trends in ALAN radiance[5]. Abrupt ALAN changes are confirmed if 14 consecutive observations are anomalous (residual > 75% change probability) in any VZA stratum. This ensures that only sustained and statistically robust deviations from the local baseline are identified. This design is well-suited to capturing sudden human-driven ALAN change events, which typically manifest as persistent and significant radiance deviations. Gradual ALAN changes are identified in segments between these breaks if the linear trend coefficient ($b_i$ in equation (3) in Supplementary Information Section 2) of the fitted 0°–60° VZA model is statistically significantly different from 0 ($P < 0.05$).

To reduce false positives arising from small fluctuations or noise, a post-processing step filters all detected changes to have a minimum magnitude threshold (1.0 nW cm$^{-2}$ sr$^{-1}$). This conservative threshold excludes minor light variations, such as day-to-day fluctuations in natural airglow. For abrupt changes, detections with change magnitude (measured in DNB radiance) below this threshold were filtered. For gradual changes, only those segments with an absolute change in DNB radiance above this threshold over the duration of the segment were considered. This dual criterion ensures that identified trends are both statistically significant and physically meaningful. Several key adaptations, including a dynamic harmonic period for high-latitude areas to handle polar day and snow impacts and a skipping-update

strategy to reduce computation, were implemented to improve model stability and reduce commission errors caused by polar-light effects, enabling robust global-scale processing. An incremental online processing framework using a 4-year moving window was adopted for continuous updates and memory management (for details, see Supplementary Information Section 2).

To quantitatively characterize detected ALAN changes, we derived a suite of pixel-based metrics, including 'change time', 'change area', 'change intensity', and 'radiance change' for abrupt, gradual and total ALAN changes, based on outputs from the adapted VZA-COLD algorithm (Extended Data Table 5). For all subsequent spatial analyses and to ensure accurate area calculations, all derived ALAN change metric maps were reprojected from their native geographic coordinates (WGS84) to a 500-m nominal resolution sinusoidal equal-area projection. The nearest-neighbour resampling method was used for this reprojection to preserve the original pixel values, particularly for categorical change information and derived intensities, while ensuring that area calculations were not distorted by variations in pixel size across different latitudes.

For each detected abrupt change, the change time was recorded as the date when the breakpoint first appeared in the fitting model. This timestamp marks the transition point between the before and after segments of the harmonic fit, often reflecting a real-world transition in lighting conditions. The change area for all pixel-based metrics was standardized to 0.2147 km$^2$, corresponding to the actual spatial footprint of the sinusoidal equal-area grid cell, enabling consistent area-based aggregation over the globe. The change intensity, described Extended Data Table 5 (equation (4)), quantifies the magnitude of an abrupt change. This was calculated to represent the net, long-term impact after the abrupt event on the stable NTL level. It was defined as the difference between the predicted overall ALAN radiance (derived from the de-seasonalized $a_i + b_i x$ terms of the 0°–60° VZA model) of the change pixel at the start date of the time segment immediately following the detected breakpoint, and the predicted overall ALAN radiance at the end date of the time segment immediately preceding the breakpoint (Supplementary Fig. 4c). A positive value for this metric indicates a step brightening in the stable light level, whereas a negative value signifies a step dimming. To quantify the total effect of an abrupt change in a spatially meaningful way, we derived the radiance change by multiplying the change intensity by the area of the pixel in Extended Data Table 5 (equation (5)). This metric expresses the sum of radiance change in ALAN and is important for scaling pixel-level transitions to broader spatial summaries.

For time segments identified as having gradual ALAN change (Supplementary Fig. 4c), the change time was marked as the dates from when the gradual change was first observed to the end date of the gradual time segment. The change area, again, was fixed at 0.2147 km$^2$. The change intensity was calculated on an annual basis, described in Extended Data Table 5 (equation (6)). This metric represents the area-averaged total change in NTL radiance attributable to the sustained gradual trend over that specific calendar year. It was calculated using the slope term from the 0°–60° VZA interval model for that particular gradual change segment (denoted as MG in equation (6) in Extended Data Table 5) multiplied by the number of days within that calendar year that belonged to that gradual change segment. This provides a measure of the intensity and direction of the gradual trend (for example, the speed of ongoing urbanization or the rate of decline in a depopulating area). The corresponding radiance change was obtained by multiplying the annual change intensity by the fixed pixel area (Extended Data Table 5, equation (7)), enabling consistent comparison and aggregation of gradual radiance changes across large regions or over multi-year periods.

To capture the cumulative impact of all detected ALAN dynamics at the pixel level, we define a set of metrics for the total change (the sum of all types of changes) as the overall alteration at a pixel resulting from the combined effects of both abrupt and gradual ALAN changes. The total change intensity metric, described in Extended Data Table 5

(equation (8)), represents the overall ALAN change intensity experienced by a pixel. It was calculated by summing the differences in radiance caused by all detected abrupt changes and all annual gradual changes occurring within a defined period (for example, for each year for time series analysis, or cumulatively over the entire 2014–2022 study period for total change mapping). This provides an integrated measure of the net impact of all ALAN dynamics on a pixel during the study period. The total radiance change metric, as described in Extended Data Table 5 (equation (9)), serves to quantify the total radiance impact of all types of ALAN changes across space.

## Accuracy assessment and unbiased area estimation

Accuracy assessment and unbiased area estimates followed established protocols for land change mapping[54,55]. An independent validation dataset was generated using a stratified random sampling design to ensure statistically robust estimations of accuracy by appropriately weighting the estimators to account for the large differences in mapped areas across classes. The sampling frame consisted of all terrestrial pixel-years (a 500-m nominal resolution sinusoidal equal-area pixel in a specific calendar year) from 2014 to 2022, totalling more than 635 million potential units. Two separate sampling schemes were implemented to target abrupt and gradual changes, respectively, as both can occur within the same year. For the abrupt change assessment, strata were defined annually as abrupt change detected or non-abrupt change (including stable and gradual change pixels). For the gradual change assessment, strata were defined as gradual change detected or non-gradual change (including stable and abrupt change pixels). Sample sizes were allocated based on mapped area proportions to achieve the accuracies of the target user of about 70% for the smaller change strata and around 95% for the larger non-change strata, with a target standard deviation of 0.005 for overall accuracy and a minimum of 200 samples per change stratum. This resulted in the selection of 2,071 sample units for the abrupt change assessment and 1,902 sample units for the gradual change assessment (Supplementary Fig. 5).

Reference data for each validation sample were generated through careful visual interpretation by trained analysts using a custom web-based application developed in Google Earth Engine. This tool facilitated consistent interpretation by showing the full VZA-stratified DNB NTL time series, corresponding high-resolution optical imagery (from Google Earth and PlanetScope) and original daily NTL image chips for each sample unit. Interpreters assigned a reference classification label (that is, Abrupt Change, Non-abrupt Change, Gradual Change, and Non-gradual Change) based on a holistic assessment of all available evidence for that specific pixel-year. During this process, analysts explicitly identified discrepancies between the map and reference labels, recording specific instances of commission errors (false positives) and omission errors (false negatives) to ensure robust accuracy estimation. For samples in which the mapped change agrees with the reference data, interpreters also qualitatively noted the potential direct causal drivers defined in Extended Data Table 4, based on the information from the remote sensing images and other open-access resources, such as the VIIRS Nightfire gas flaring data from SkyTruth[56], conflict data from UCDP (Uppsala Conflict Data Program), earthquake data from CrisisWatch, hurricane records from NOAA (National Oceanic and Atmospheric Administration), social media (for example, X and Weibo), and relevant news and reports. To ensure objectivity in this driver attribution, we implemented a strict quality control protocol: each sample was independently interpreted by two trained analysts. Any discrepancies between their driver labels were flagged and adjudicated by a third, senior analyst to produce the final consensus label.

Based on a comparison of map labels to reference labels, confusion matrices were constructed separately for the abrupt and gradual change maps (Extended Data Table 1). From these matrices, unbiased estimators of overall accuracy, user accuracy (a measure of commission error) and producer accuracy (a measure of omission error) were calculated with

95% CIs, appropriately weighted by the area proportions of each stratum to avoid bias from large stable areas[54,57]. To obtain statistically robust and unbiased estimates of the cumulative area experiencing different types of ALAN change (globally and regionally), we applied area-weighted estimators based on the reference samples[54]. This standard method leverages the stratified validation sample to correct map-based pixel counts for classification errors, providing unbiased area estimates with associated 95% CIs (Extended Data Table 3). Note that apart from these unbiased cumulative area estimates, all other area-related spatial analyses in this study rely on direct mapped pixel counts.

### Spatial and temporal analysis

For analysing broad spatial patterns and for visualization purposes, the 15-arc-second pixel-level ALAN change metrics were aggregated into regular linear latitude/longitude grid cells. Two grid resolutions were primarily used: 0.5° and 2°. Within each of these grid cells, summary statistics such as the sum of change area, the area-averaged change intensity and the sum of radiance change were calculated (see definitions in Extended Data Table 5). This gridded aggregation provided a uniform spatial framework that facilitated the visual analysis of broad spatial patterns across the globe (0.5°) and the calculation of zonal statistics (with 2°), such as those used for creating the latitudinal and longitudinal profile plots (Fig. 3 and Extended Data Fig. 2). Moreover, for analyses tied to geopolitical or socioeconomic contexts, the equal-area 500-m nominal resolution sinusoidal pixel-level results were aggregated based on administrative boundaries (continents, countries, and territories) obtained from the World Bank dataset. Total change area, area-averaged change intensity and total radiance change were calculated for each administrative unit.

To evaluate how the extent and intensity of ALAN change evolved over the 2014–2022 study period, we performed temporal trend analysis on the annually aggregated statistics. This was done at global and, where data permitted, country and territory levels. The non-parametric Theil–Sen regression method[58–60] was used to estimate the slope (specifically, the median slope) of the trend in annual ALAN change over the 9-year period. Trends were considered to be statistically significant if the $P$-value from the Mann–Kendall test was less than 0.05. These trend tests were applied separately to metrics for net ALAN change (sum of brightening and dimming), brightening ALAN change and dimming ALAN change (Fig. 4).

Although the actual change values (for example, change area in km$^2$, change intensity in nW cm$^{-2}$ sr$^{-1}$) provide direct measures of alteration, understanding the rate or relative change compared with the initial state of illumination is often crucial for contextualizing the impact of ALAN changes, especially when comparing regions with vastly different baseline light levels. Using simple annual composites from the start year (that is, 2014) as a baseline for calculating relative change can be misleading, as these empirical data can be affected by short-term variability, seasonality or even undetected early changes that occurred before the model initialization was complete. Therefore, to establish a more robust and stable baseline representing the initial NTL conditions, we generated synthetic baseline NTL radiance values and area for 1 January 2014 (Extended Data Table 2). This was achieved by using the overall trend terms of the harmonic model that was fitted to the 0°–60° VZA interval data, primarily based on the 2013 observations used for model initialization ($a_i + b_i x$ in Supplementary Information Section 2, equation (3)). We defined the global lit area of the 2014 baseline (or initial global lit area) as all regions in which this synthetic baseline radiance exceeded 1.0 nW cm$^{-2}$ sr$^{-1}$. This calculation effectively removes the influence of seasonality captured by the harmonic terms and represents the de-seasonalized, stable baseline radiance that was predicted by the initial model fit, thus being free from the influence of any ALAN changes detected after the model was initialized. These synthetic baseline radiance values provide a consistent and unbiased representation of initial lighting conditions across all pixels. Relative changes observed over the study period (for example, percentage radiance change in ALAN) were then calculated by comparing the observed cumulative radiance change to these synthetic baseline radiance values (Fig. 3e).

## Data availability

The open-source data include the following: the VIIRS/NPP Gap-Filled Lunar BRDF-Adjusted Nighttime Lights Daily L3 Global 500 m Linear Latitude/Longitude Grid (VNP46A2) at https://ladsweb.modaps.eosdis.nasa.gov, the NASA Black Marble Collection 1 VIIRS/NPP Daily Gridded Day Night Band 500 m Linear Lat Lon Grid Night (VNP46A1) at https://ladsweb.modaps.eosdis.nasa.gov, the 2014 MODIS/Terra Land Water Mask derived from MODIS and SRTM L3 Global 250 m SIN Grid Collection 6.1 (MOD44W) at https://www.earthdata.nasa.gov, the World Bank-approved administrative boundaries at https://datacatalog.worldbank.org. The change dataset generated in this study is publicly available at https://doi.org/10.5281/zenodo.18264642.

## Code availability

The global ALAN change dataset and analyses were produced with custom code using MATLAB 2022b and Python 3.10, which are available at Zenodo[61] (https://doi.org/10.5281/zenodo.18264642).

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

**Acknowledgements** This work was supported by the Terra, Aqua, Suomi-NPP and NOAA-20 programs of NASA (grant 80NSSC22K0199) and the Remote Sensing Theory for Earth Science program of NASA (grant 80NSSC20K1748). C.C.M.K. was supported by the New Earth Observation Mission Ideas program of ESA (contract 4000139244/22/NL) and by the DFG (grant 545098235). Y.Y. was partially supported by the NASA ACRES project (grant 80NSSC23M0034). The computational work for this project was conducted using resources provided by the Storrs High-Performance Computing (HPC) cluster. We extend our gratitude to the UConn Storrs HPC and its team for their resources and support, which aided in achieving these results. We acknowledge the use of AI-based language tools to improve the clarity and readability of the manuscript, and all content was reviewed and approved by the authors. Any use of trade, firm or product names is for descriptive purposes only and does not imply endorsement by the US government.

**Author contributions** T.L., Z.Z. and Z.W. conceptualized the study; T.L., Z.Z. and Z.W. devised the methodology; T.L. was involved in production; T.L., S.Q., M.C., F.H., A.G., K. Song, J.W.S. and X.Y. performed the product validation; T.L., Z.Z., Z.W., C.C.M.K., M.O.R. and K.C. Seto conducted the formal analysis; T.L. and Z.Z. wrote the original draft; T.L., Z.Z., Z.W., C.C.M.K., M.O.R., K.C. Seto, Y.Y., S.Q., T.K., M.F., X.C., T.H.M., C.D.R., X.T., M.C., F.H., A.G., K. Song, J.W.S., X.Y., V.L.K. and C.D. reviewed and edited the paper; Z.Z. and Z.W. helped with funding acquisition.

**Competing interests** The authors declare no competing interests.

**Additional information**
**Correspondence and requests for materials** should be addressed to Tian Li or Zhe Zhu.

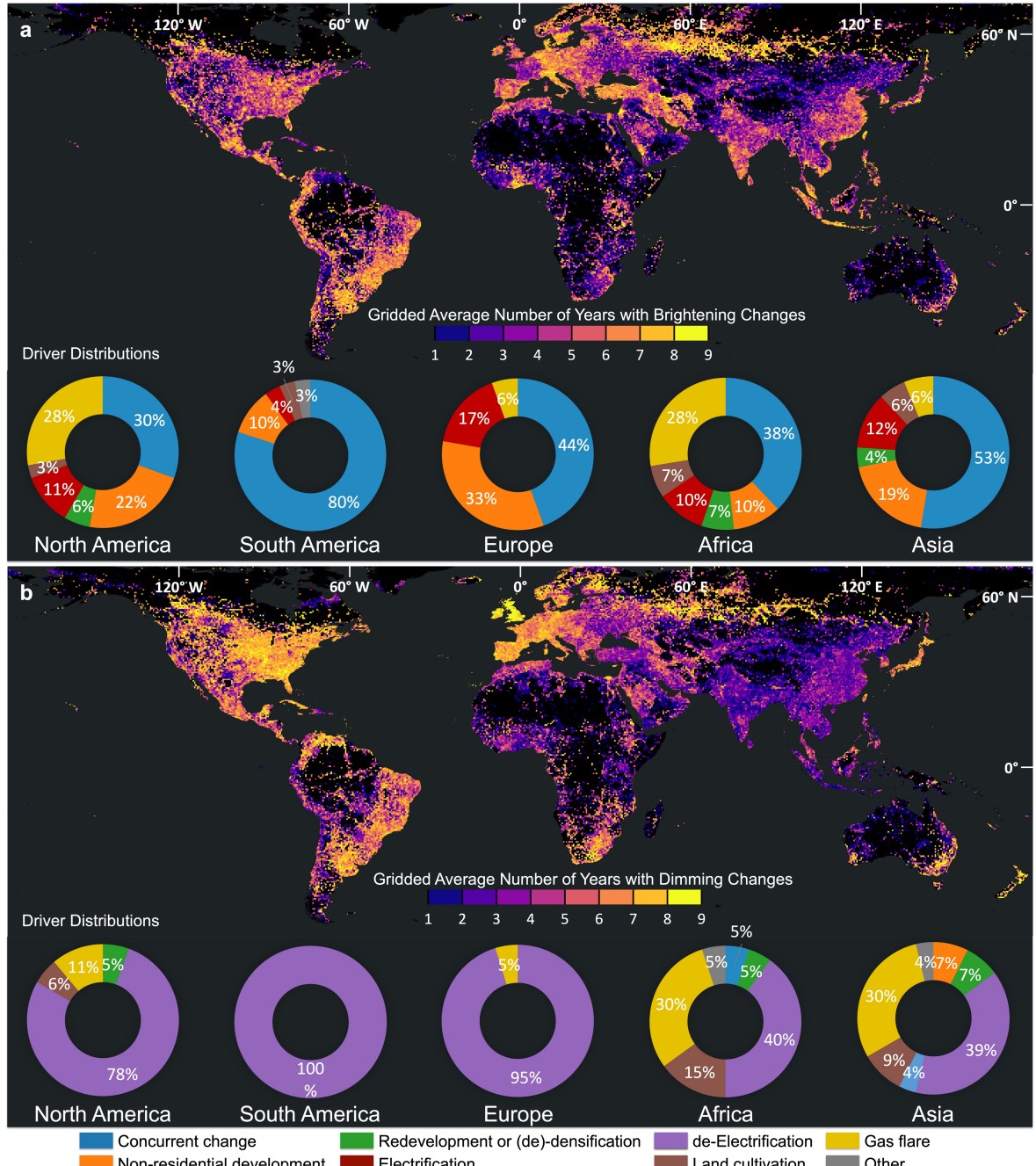

**Extended Data Fig. 1 | Temporal frequency and estimated drivers of global ALAN brightening and dimming from 2014 to 2022. a,b**, Maps show the gridded average number of years (0.5° × 0.5° grid cell) experiencing brightening (**a**) and dimming (**b**). Donut charts showing the sample-estimated (Extended Data Table 1) proportions of causal drivers associated with brightening and dimming across continents. Oceania is excluded due to limited changes and insufficient sample units. Basemaps in all panels are adapted from the World Continents Layer from Esri.

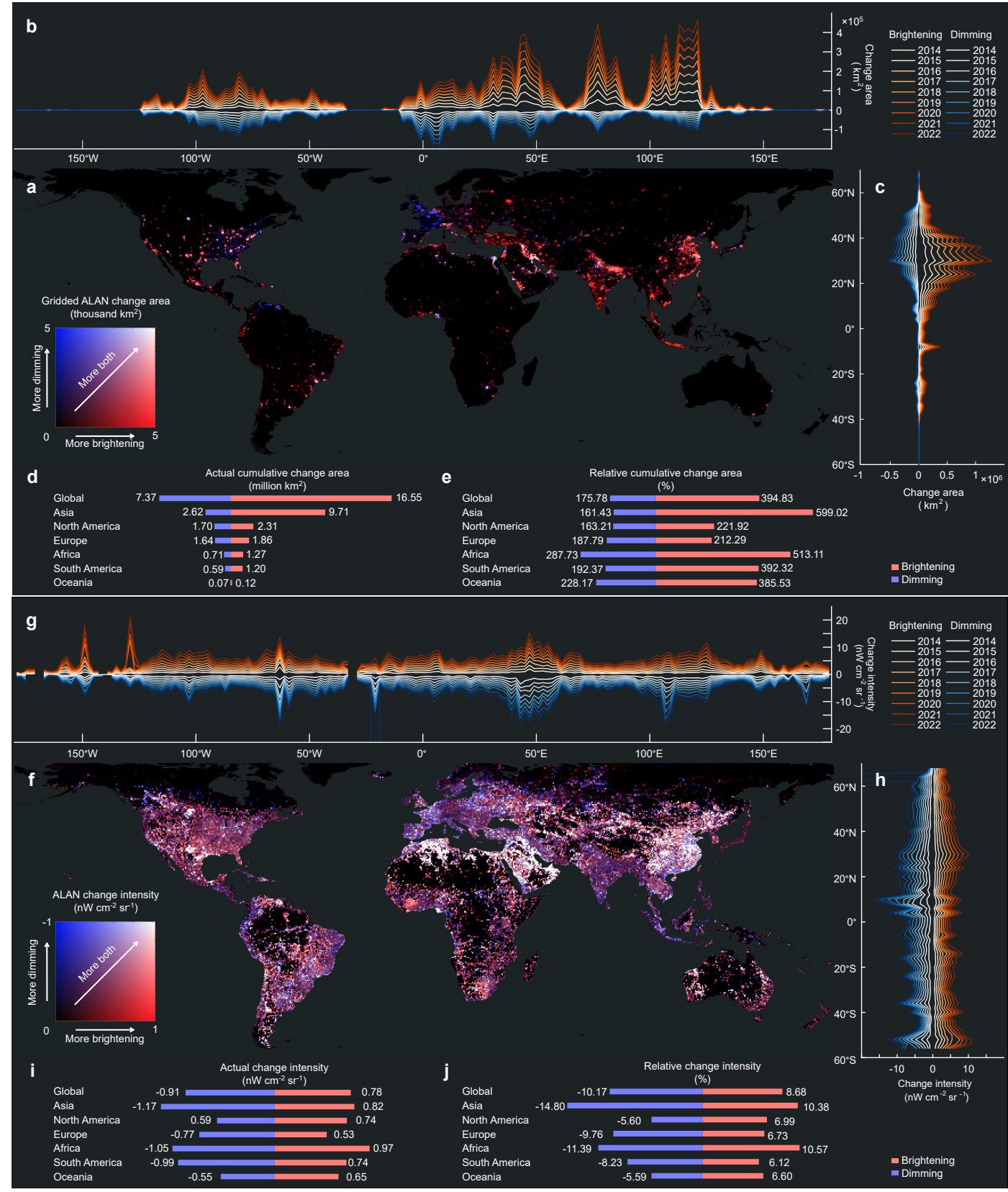

**Extended Data Fig. 2** | See next page for caption.

**Extended Data Fig. 2 | Global patterns of ALAN change area and change intensity from 2014 to 2022.** Panels **a**–**e** focus on ALAN change area: **a**, Gridded heat map shows the cumulative sum of ALAN change areas. Values represent the cumulative sum of all brightening change areas (from black to red) and the cumulative sum of all dimming change areas (from black to blue) within each 0.5° × 0.5° grid cell. Brighter colours mean larger cumulative change areas in both directions, and darker colours mean the opposite. **b**,**c**, Longitudinal (**b**) and latitudinal (**c**) cumulative profile plots of the yearly sum of brightening (red/orange tones) and the yearly sum of dimming (blue/cyan tones) in every 2° step. **d**, Actual ALAN cumulative change area summed at global and continental levels. **e**, Relative ALAN cumulative change areas by the end of 2022, compared with the 2014 baseline, at global and continental levels. The red bars in **d** and **e** represent the brightening changes, and the blue bars represent the dimming changes. Panels **f**–**h** focus on ALAN change intensity: **f**, Gridded heat map shows change intensity. Values represent the brightening change intensity (from black to red) and the dimming change intensity (from black to blue) within each 0.5° × 0.5° grid cell. Brighter colours mean larger change intensities in both directions, and daker colours mean the opposite. **g**,**h**, Longitudinal (**g**) and latitudinal (**h**) cumulative profile plots of the yearly brightening and the yearly dimming ALAN change intensity in every 2° step. **i**, Actual ALAN change intensity at global and continental levels. **j**, Relative ALAN change intensity by the end of 2022, compared with the 2014 baseline, at global and continental levels. The red bars in **i** and **j** represent the brightening changes, and the blue bars represent the dimming changes. Basemaps in **a** and **f** are adapted from the World Continents Layer from Esri.

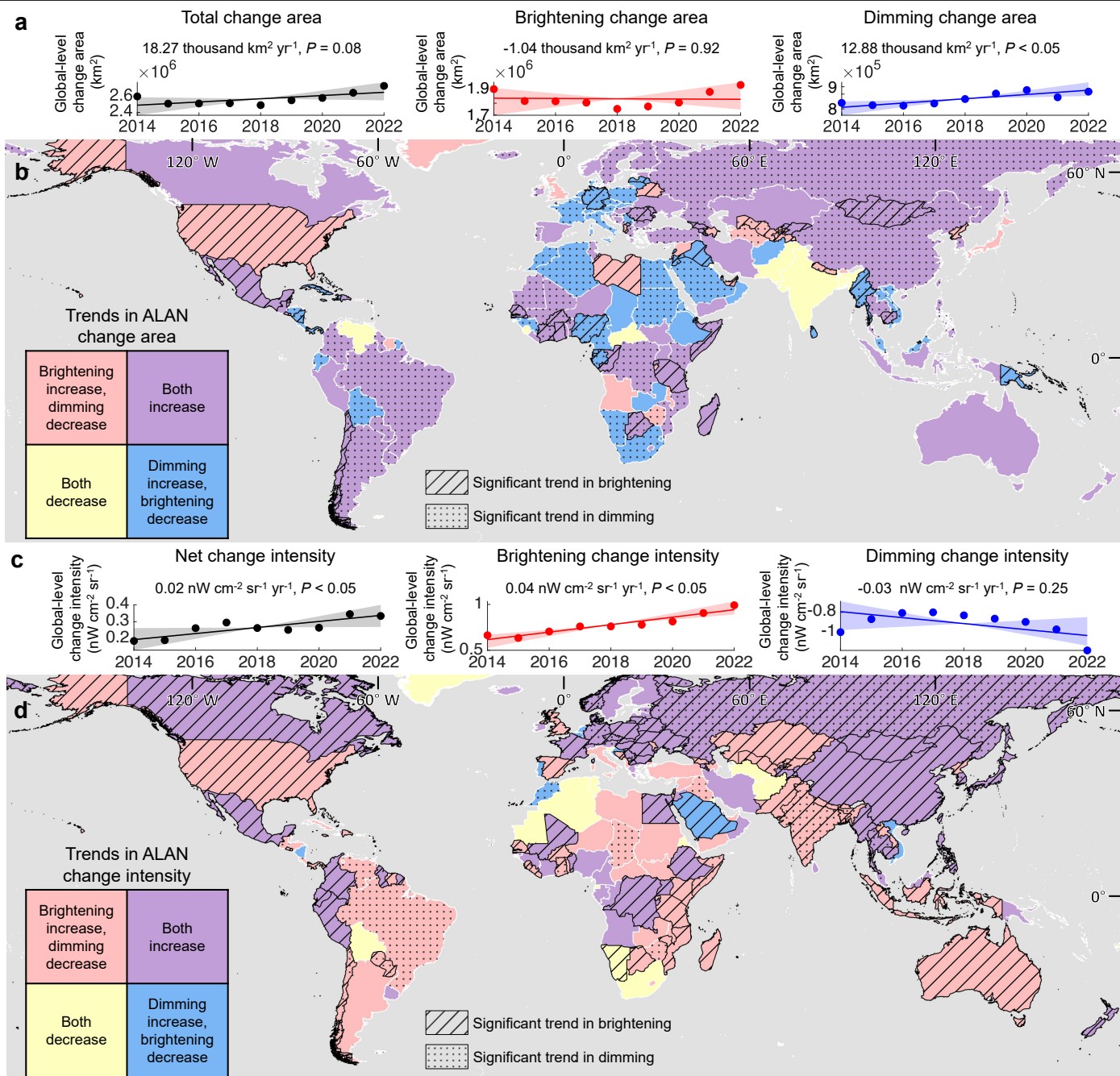

**Extended Data Fig. 3 | Global and country/territory-level temporal trends of ALAN change area and change intensity (2014–2022). a**, Line plots showing the global trends in the annual area experiencing ALAN change: total change area (left); brightening change area (middle); and dimming change area (right). **b**, Bivariate map classifying country/territory level trends in change areas. Countries/territories are categorized based on whether their annual area of brightening is increasing vs. decreasing, crossed with whether their annual area of dimming is increasing vs. decreasing. **c**, Line plots showing the global trends in the annual ALAN change intensity: average net change intensity (left); average brightening change intensity (middle); and average dimming change intensity (right). **d**, Bivariate map classifying country/territory level trends in change intensity. Countries/territories are categorized based on whether their annual brightening change intensity is increasing vs. decreasing, crossed with whether their annual dimming change intensity is increasing vs decreasing. For line plots (**a**,**c**): the dots represent the global annual ALAN change area or change intensity from 2014 to 2022, the solid lines show the overall Sen's slope trends, and the shaded areas represent the 95% confidence interval of the estimated trends. For bivariate maps (**b**,**d**): the underlying map colours represent the combined topology of trends, the stripe texture indicates countries/territories with statistically significant trends in brightening changes (either change area or change intensity, corresponding to the panel), and the dot texture indicates countries/territories with statistically significant trends in dimming changes. Significance is determined by the Mann-Kendall test (p < 0.05) on the Sen-slope. Basemaps in **b** and **d** are adapted from the World Continents Layer from Esri.

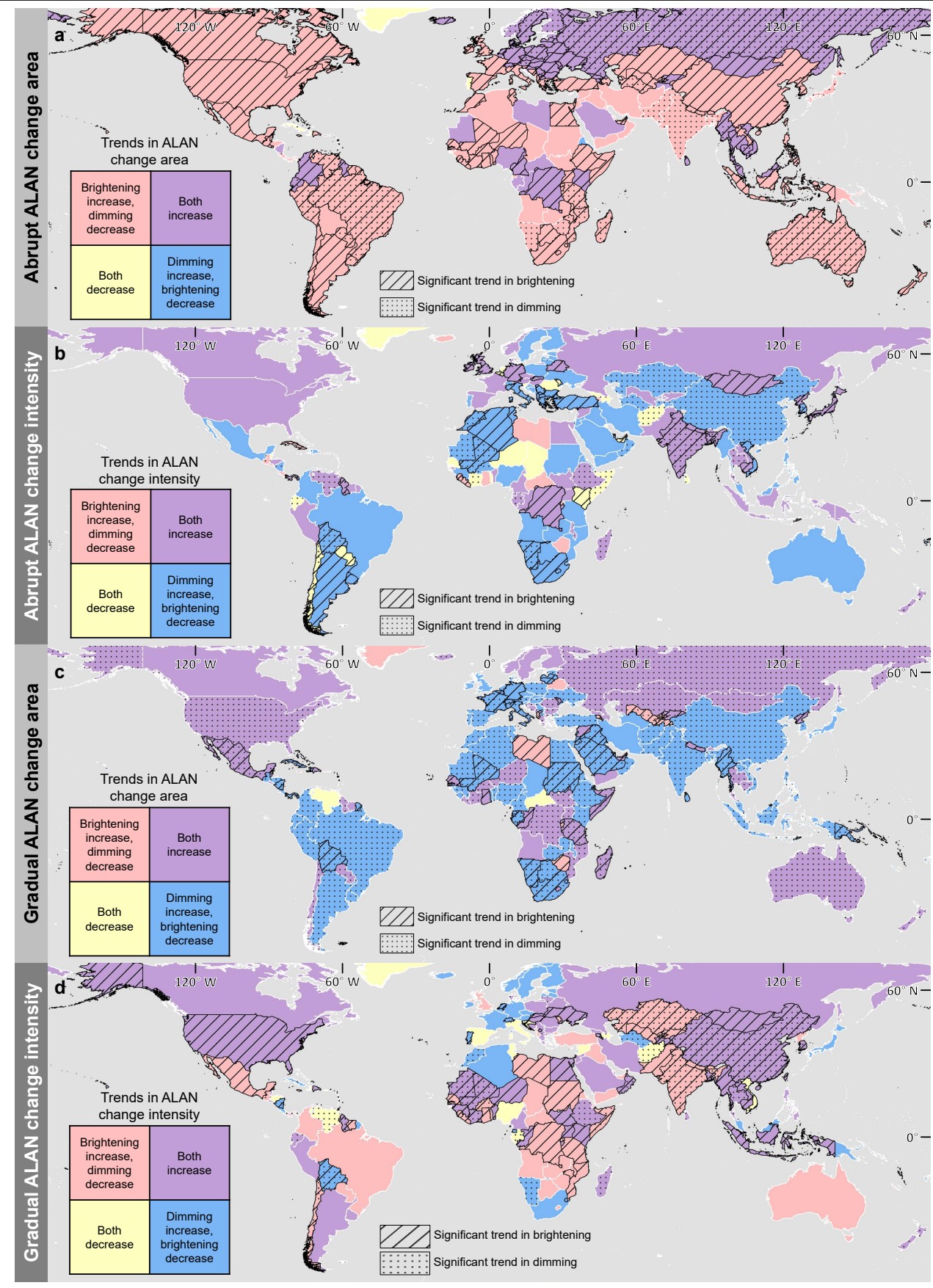

**Extended Data Fig. 4** | See next page for caption.

**Extended Data Fig. 4 | Temporal trend of country/territory level ALAN change area and change intensity (2014–2022).** This figure classifies countries/territories based on the interaction of statistically significant (Sen-slope, Mann-Kendall p < 0.05) temporal trends in their annual ALAN change metrics. **a**–**b**. Bivariate maps showing the typology of trends in annual abrupt brightening and dimming change area (**a**) and change intensity (**b**), respectively. **c**–**d**. Bivariate maps showing the typology of trend in gradual ALAN brightening and dimming change area (**c**) and change intensity (**d**), respectively. Basemaps in all panels are adapted from the World Continents Layer from Esri.

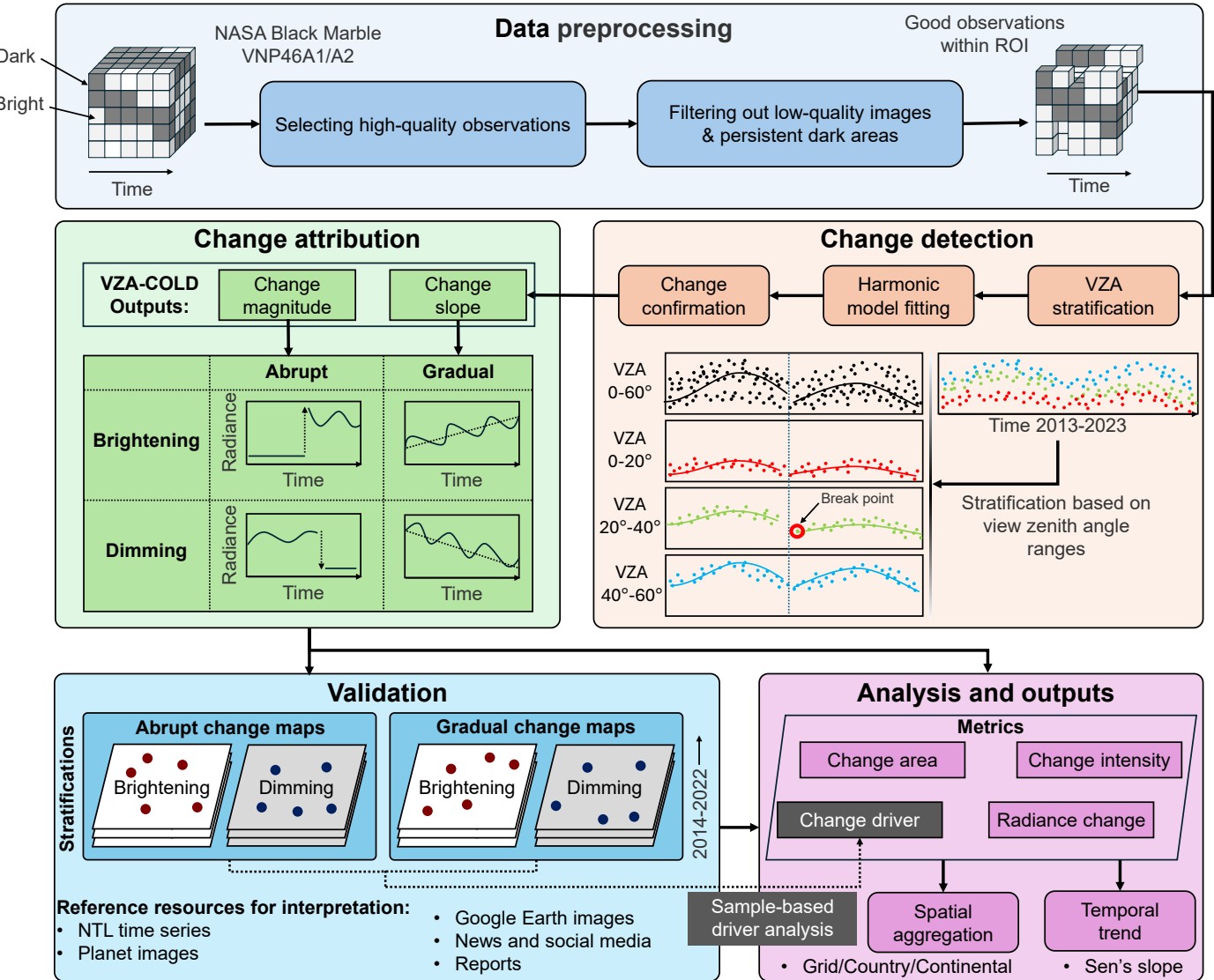

**Extended Data Fig. 5 | Flowchart of this study methodology.** The input data consists of the NASA Black Marble Collection 1 product. The selection of high-quality observations involved filtering out clouds and snow (dilated by five pixels) and excluding water pixels. Break points are detected if a change is observed in any stratified viewing zenith angle range (e.g., 20°–40°); identified break points are then applied to the time series models across all ranges. The change metrics are derived from the all angle (0–60°) observations to ensure consistency. ROI: Region Of Interest, defined as the global analyzed nighttime light areas from 70°N to 60°S. VZA-COLD: Viewing Zenith Angle stratified COntinuous monitoring of Land Disturbance.

**Extended Data Table 1 | Accuracy assessment of global ALAN change maps (2014–2022)**

| Abrupt Change Accuracy | | Reference | |
|---|---|---|---|
| | | Abrupt | Non-Abrupt |
| **Confusion Matrix of Sample Counts (pixel)** | | | |
| **Map** | Abrupt | 169 | 31 |
| | Non-abrupt | 3 | 1,868 |
| **Confusion Matrix of Area Proportions (%)** | | | |
| **Map** | Abrupt | 1.35 | 0.25 |
| | Non-abrupt | 0.16 | 98.25 |
| **Accuracy Estimates (%)** | | | |
| **User's** | | 84.50 ± 5.03 | 99.84 ± 0.18 |
| **Producer's** | | 89.52 ± 10.63 | 99.75 ± 0.08 |
| **Overall** | | 99.59 ± 0.20 | |

| Gradual Change Accuracy | | Reference | |
|---|---|---|---|
| | | Gradual | Non-Gradual |
| **Confusion Matrix of Sample Counts (pixel)** | | | |
| **Map** | Gradual | 262 | 48 |
| | Non-gradual | 4 | 1588 |
| **Confusion Matrix of Area Proportions (%)** | | | |
| **Map** | Gradual | 13.75 | 2.52 |
| | Non-gradual | 0.21 | 83.52 |
| **Accuracy Estimates (%)** | | | |
| **User's** | | 84.52 ± 4.03 | 99.75 ± 0.25 |
| **Producer's** | | 98.49 ± 1.46 | 97.07 ± 0.74 |
| **Overall** | | 97.27 ± 0.69 | |

Confusion matrices and accuracy estimate for the abrupt and gradual change maps. Non-abrupt refers to pixels that were stable or experienced only gradual change, while non-gradual refers to pixels that were stable or experienced only abrupt change within the analyzed ALAN region (Supplementary Fig. 3). The uncertainty ± indicated the margin of error of 95% confidence interval. The distribution of these validation sample units is shown in Supplementary Fig. 5.

**Extended Data Table 2 | Baseline ALAN conditions and the ever-changed areas (2014–2022)**

| Region | Initial Area (km²) | Change Rate (%) | Ever-Changed Area (km²) | Ever-Changed Area (Abrupt) (km²) | Ever-Changed Area (Gradual) (km²) |
|---|---|---|---|---|---|
| Global | 4,190,898 | 83.73% | 3,509,097 | 1,715,400 | 3,004,940 |
| Africa | 247,280 | 119.35% | 295,132 | 171,071 | 248,458 |
| Asia | 1,621,582 | 115.25% | 1,868,829 | 1,085,842 | 1,579,832 |
| Europe | 875,825 | 56.59% | 495,649 | 165,093 | 434,927 |
| N. America | 1,039,439 | 53.82% | 559,473 | 194,823 | 486,020 |
| Oceania | 30,728 | 94.39% | 29,003 | 12,590 | 22,998 |
| S. America | 305,451 | 79.17% | 241,836 | 79,946 | 215,609 |
| China | 407,388 | 129.57% | 527,839 | 329,987 | 433,527 |
| USA | 792,139 | 54.15% | 428,937 | 151,157 | 368,664 |
| India | 363,831 | 118.96% | 432,828 | 283,536 | 357,332 |
| Brazil | 148,082 | 77.82% | 115,240 | 36,262 | 103,817 |
| Iran | 108,848 | 98.36% | 107,063 | 53,136 | 98,076 |

The table presents initial baseline conditions as of January 1, 2014, the area of pixels experiencing at least one change, and the area experiencing at least one abrupt (or gradual) change between 2014 and 2022. The top five countries/territories (last five rows) are ranked based on their net ALAN radiance change. N.: North. S.: South.

**Extended Data Table 3 | Unbiased cumulative ALAN change area from 2014 to 2022**

| Region | | Cumulative Area of ALAN Change (million km²) | |
|---|---|---|---|
| | | Abrupt Change | Gradual Change |
| **Global** | | 2.05 [1.79, 2.32] | 19.04 [18.10, 19.98] |
| **Continent** | Africa | 0.28 [0.14, 0.41] | 1.77 [1.48, 2.06] |
| | Asia | 1.22 [1.05, 1.39] | 9.73 [9.14, 10.31] |
| | Europe | 0.17 [0.13, 0.20] | 2.51 [2.02, 3.00] |
| | North America | 0.21 [0.18, 0.24] | 3.13 [2.70, 3.56] |
| | Oceania | *Not Estimated* | 0.18 [0.18, 0.18] |
| | South America | 0.15 [0.01, 0.30] | 1.50 [1.30, 1.70] |
| **Country/Territory** | China | 0.37 [0.34, 0.41] | 2.54 [2.17, 2.91] |
| | India | 0.23 [0.18, 0.29] | 2.01 [1.76, 2.26] |
| | USA | 0.16 [0.13, 0.19] | 2.47 [2.12, 2.82] |
| | Brazil | 0.10 [-0.03, 0.24] | 0.74 [0.62, 0.86] |
| | Iran | 0.08 [0.06, 0.10] | 0.69 [0.58, 0.80] |

Cumulative ALAN change areas at global, continental, and top five countries/territories were estimated. The same location is counted multiple times if it experiences multiple changes. These unbiased cumulative area estimates and their 95% confidence intervals were derived from the stratified validation sample in Extended Data Table 1. The abrupt cumulative change area was *not estimated* in Oceania due to a small proportion of change and insufficient validation change sample unit ($n \le 1$) in this subregion.

**Extended Data Table 4 | Definitions of major drivers of ALAN change**

| Driver Category | Definition |
|---|---|
| **Concurrent Change** | Commonly associated with "urbanization," this occurs where population, land use, and electric infrastructure change simultaneously. It typically reflects a significant transformation in residential areas. |
| **Non-residential Development** | Occurs when built-up land and electricity infrastructure are expanding, but population numbers are stable or lag behind. This is characteristic of non-residential development, including commercial, industrial, or public infrastructure. |
| **Redevelopment or (de)-Densification** | Includes areas with population change but relatively stable land use. Redevelopment involves simultaneous changes to population and electricity infrastructure (e.g., urban revitalization, property abandonment, remodeling). (de)-Densification involves population change within existing structures without significant physical expansion or vertical growth. |
| **Electrification** | Occurs in regions with stable land cover and population but substantial growth in electric infrastructure, such as new grid expansion. |
| **de-Electrification** | Occurs in regions with stable land cover and population but a substantial decline in ALAN, often associated with energy-saving measures like LED streetlight renovation or declines in energy access. |
| **Disaster** | An event causing severe danger and significant destruction, including natural hazards (e.g., hurricanes, earthquakes, flooding) or human-caused events (e.g., armed conflicts, epidemics) that disrupt lighting infrastructure. |
| **Land Cultivation** | Occurs when land cover is changing in isolation. This category includes sites of seasonal agriculture, mining, or other excavation industries. |
| **Gas Flare** | The combustion of natural gas, typically associated with oil and gas extraction sites, refineries, or processing plants. |
| **Other** | Any change event where the driver does not fit the defined categories above or could not be confidently interpreted from the available data. |

This table defines the categories used for the visual interpretation of the direct, proximate drivers of the ALAN change validation sample. The driver typology is adapted from the urban change framework proposed by Stokes and Seto (2019)[29].

**Extended Data Table 5 | Definitions of pixel-based ALAN change metrics**

| Change Type | Change Metrics | Equation | | Definition |
|---|---|---|---|---|
| **Abrupt Change** | Change Time | - | | Date (Year, Day-of-Year) when the abrupt change was detected. |
| | Change Area | $\mathrm{ChgArea_{Abr}} = 0.2147\ km^2$ | | The fixed actual spatial extent each "500-m" nominal resolution equal-area sinusoidal grid abrupt change pixel represent. |
| | Change Intensity | $\mathrm{ChgInt_{Abr}} = (a_{MA} + b_{MA}\mathrm{t\_start}_{MA}) - (a_{MB} + b_{MB}\mathrm{t\_end}_{MB})$ | (4) | Area-averaged ALAN radiance change between the beginning of the after-abrupt-change segment and the end of the before-abrupt-change segment. |
| | Radiance Change | $\mathrm{ChgRad_{Abr}} = \mathrm{ChgInt_{Abr}} \times \mathrm{ChgArea_{Abr}}$ | (5) | Overall ALAN radiance change for a given area due to abrupt change. |
| **Gradual Change** | Change Time | - | | Start and end dates of the gradual change time segment. |
| | Change Area | $\mathrm{ChgArea_{Grd}} = 0.2147\ km^2$ | | The fixed actual spatial extent each "500-m" nominal resolution equal-area sinusoidal grid gradual change pixel represent. |
| | Change Intensity | $\mathrm{ChgInt_{Grd}} = b_{MG}(\mathrm{t\_end}_{MG} - \mathrm{t\_start}_{MG})$ | (6) | Area-averaged ALAN radiance change for each year due to gradual change. |
| | Radiance Change | $\mathrm{ChgRad_{Grd}} = \mathrm{ChgInt_{Grd}} \times \mathrm{ChgArea_{Grd}}$ | (7) | Overall ALAN radiance change for a given area due to gradual change. |
| **Total Change** | Change Area | $\mathrm{ChgArea_{Ttl}} = 0.2147\ km^2$ | | The fixed actual spatial extent each "500-m" nominal resolution equal-area sinusoidal grid abrupt or gradual change pixel represent. |
| | Change Intensity | $\mathrm{ChgInt_{Ttl}} = \mathrm{ChgInt_{Abr}} + \mathrm{ChgInt_{Grd}}$ | (8) | Total area-averaged ALAN radiance change, calculated as the sum of abrupt change intensity and gradual change intensity. |
| | Radiance Change | $\mathrm{ChgRad_{Ttl}} = \mathrm{ChgRad_{Abr}} + \mathrm{ChgRad_{Grd}}$ | (9) | Total ALAN radiance change for a given area due to abrupt and gradual changes |

where, Abr means the abrupt ALAN change, Grd means the gradual ALAN change, Ttl means the sum of abrupt and gradual ALAN change, a and b are the constant and slope term coefficients in equation (3), t_start is the start time of a time segment, t_end is the end time of a time segment, MA is the prediction model of the 0–60 VZA interval for the time segment after the abrupt ALAN change, MB is the prediction model of the 0–60 VZA interval for the time segment before the abrupt ALAN change, MG is the prediction model of the 0–60 VZA interval with significant (p<0.05) gradual ALAN change identified.