## [Peer Review file · Nature]

Satellite Imagery Reveals Increasing Volatility in Human Nighttime Activity

Corresponding Author: Dr Zhe Zhu

Version 0:

Reviewer comments:

Referee #1

(Remarks to the Author)

NTL = night time lights

L35 = line 35 in the manuscript etc.

A. Summary of the key results

This paper is about an interesting and novel global analysis of night lights, at a high temporal resolution. The core method, to identify both abrupt and gradual NTL changes, is rational and tested. The main interesting element of the manuscript is the focus on the “bidirectional dynamism”, the fact that with NTL the seasonality of urban areas can be captured as well as the impact of several ephemeral and ad hoc events (natural and anthropogenic disasters, pandemics, conflicts etc). NTL are uniquely correlated with socioeconomic activity and urban areas are characterized by constant shifting patterns of activity, even within the same day (e.g. commuting). Classic (day) optical remote sensing can not capture these trends. NTL fill this gap. The paper clearly shows that.

B. Originality and significance: if not novel, please include reference

The use of daily NTL data on a global scale is novel and significant. The exploitation of NTL fluctuations to describe development trends is original.

C. Data & methodology: validity of approach, quality of data, quality of presentation

The data and methods are valid and of good quality. The quality of presentation is overall adequate.

L116 Since the sinusoidal projection (equal area) is already used by the authors, perhaps it would be a better option to use it and make the global maps in the paper, as the geographic projection WGS84 heavily distorts areas and it not a good choice for global maps.

Figure 5.d, in the x-axes, the second ‘Jan2020’ should perhaps be Jan2021?

D. Appropriate use of statistics and treatment of uncertainties

Statistics and uncertainties are overall well addressed.

L860 k-hat statistic has also to be reported (not just overall accuracy) given that the two classes are very different in sample size.

E. Conclusions: robustness, validity, reliability

While the strong point of this paper is the use of high temporal resolution (daily) NTL and the capacity to capture light fluctuations in short time intervals, I believe that the manuscript will greatly gain by explaining more what is the actual added value of monitoring high temporal frequency fluctuations globally, with daily data. What exactly do coexisting brightening and dimming trends show for an area? What is the benefit to know that two regions both exhibit overall increasing NTL trends but one of the two shows more temporal fluctuation than the other? It is stated in the text but I think a more focused

discussion will improve the objective.

F. Suggested improvements: experiments, data for possible revision

Two main points to consider:

1) L35, L59-61, "the prevailing assumption of near-universal growth in light radiance" I don't think we can say that there is such a unanimous assumption in the literature. Elvidge et al. for example, identify 68 countries with declining, stable, erratic lights (i.e. not growing). This does not sound very different compared to the findings in L167-168. Please consider rephrasing.

- Elvidge C et al., 2011, National Trends in Satellite Observed Lighting: 1992-2009, AGU Fall Meeting Abstracts, <https://ui.adsabs.harvard.edu/abs/2011AGUFMGC32C..03E>

2) L79-80, L158, L355, L386-7, L782-785 etc "Focusing only on the net change year-on-year, we miss the crucial temporal ...variation..". In several instances, throughout the text, it is almost implied that NTL temporal resolution moved from annual to daily, in one step. But in fact, the temporal NTL resolution was initially annual (DMSP/OLS), then it became monthly (VIIRS available since 2012) and more recently it became daily (black marble). The monthly data needs to receive more attention in the manuscript, because it is already available for several years, and solve the seasonality problem for many applications that do not need the daily resolution. Monthly data is a good basis for most analysis within the year, accounting for phenology etc. And several such applications have appeared using monthly data to examine virus spread, population seasonality etc. The clear acknowledgment of the monthly VIIRS data will also help the authors to better focus on the benefit of the daily data. What can daily data do that monthly can not? Also note, that monthly averages are in fact used in some parts of the manuscript (L391, L637-642). Again the benefit of using daily data (instead or monthly) should become clearer.

https://developers.google.com/earth-engine/datasets/catalog/NOAA_VIIRS_DNB_MONTHLY_V1_VCMSLCFG

Some minor points:

3) L81-82 "This oversight [using only annual data] limits our ability to truly link cause and effect..." Similarly to spatial resolution, the highest technically available temporal resolution is not by default best for any application. Please consider better explaining this sentence.

4) L93 It would be useful to briefly summarize sources and magnitude of noise here.

5) L169 Please check author names.

6) L180 Please check "oremerging"

7) L225 According to EUROSTAT the population at the EC 27 level (The European Union's 27 member states) is not declining. The population in France, the UK and in the Netherlands is also not declining. In fact, it is slightly increasing (https://ec.europa.eu/eurostat/databrowser/view/tps00001/default/table?lang=en&category=t_demo.t_demo_pop)

8) L226 Please consider moving Venezuela to an other paragraph, as the discussion here is for Europe.

9) L268-283 Could the discussion become more specific to better show the causes of NTL fluctuation?. How urban density and evolving land use patterns affect NTL trends?

10) Figure 4.b it is a bit surprising to see e.g. Italy and France in the same class with Syria and Yemen, all considered high baseline nations, now experiencing dimming. I think that the dimming in these countries is attributed to very different reasons (policy/technology advancements vs extensive conflicts). See also my next comment below about the typology.

11) L880 The typology contains the class "(de)-electrification" for stable regions, with both growing or declining trends. An interesting finding of the paper is that some developed counties show declining trends, attributed to a policy and/or technological shift, adopting LED tech etc (L220-221, L224, L233-4, L417-421). Therefore, in my opinion, it would be interesting to split this class in two, and have a separate class as stable+declining. The new class would be interesting as it is a totally different case, compared to de-electrification due to crisis and disasters. More countries will probably be added to this class in the future. Also it supports the point made in the manuscript for a need to rethink the association of NTL with socioeconomic variables (L417). This is the main point to rethink in my view. A future negative correlation of development with NTL. France in Figure 5.c is a clear example.

12) L870 the % change would also be useful. Possibly by removing the radiance column that is not further used to show change, to make space in the table.

G. References: appropriate credit to previous work?

Overall, references are appropriate.

L421 the use of this reference needs a bit more explaining [46]. Is it the correct reference?

H. Clarity and context: lucidity of abstract/summary, appropriateness of abstract, introduction and conclusions

Abstract, introduction and conclusions are appropriate and informative.

(Remarks on code availability)

I reviewed but did not run the code. The code provides sufficient information to reproduce it and is well documented and valuable.

Referee #2

(Remarks to the Author)

By adding the time dimension, this paper represents a significant advance in understanding anthropogenic night sky light. The authors state they are challenging the prevailing concept of near-universal ALAN growth. Further this work if widely disseminated and understood will be a powerful resource to inform the worldwide effort to preserve the natural night sky. For these and other reasons this paper should be held to a very high standard.

I would recommend publication provided some important issues are addressed in a revision.

Given the definition of artificial, it is essential that ALAN be measured relative to or at least thoughtfully compared to the brightness of the natural night sky. The intensity of ALAN ranges from a maximum, usually in large cities, to a value of near zero in remote locations where natural night sky brightness variations dominate. The authors designation of "Persistently dark regions" are presumably such locations. These areas of natural night sky brightness should be included in the analysis. Typically the brightness of the natural airglow can vary by 40% or more even during the course of a single more or less normal night. In areas of slight to moderate ALAN, variations in the brightness of the natural night sky print through. Failure to account for this situation will result in erroneous interpretations of the data.

The authors present specific examples that ALAN provides a unique view of many aspects of human activity making their results potentially useful to a wide variety of research fields. If this article is being submitted to Nature it should be written to communicate with researchers in a wide variety of fields. Words alone may not be sufficient. Several flow chart/organization types of graphics could help the serious reader to visualize and utilize the concepts and procedures described in this paper. Given the resolution of the figures in the manuscript, the authors should consider adding an extensive online interactive component so readers can ask their own questions and see how the data and analysis supports the conclusions presented in the text.

An interactive light pollution map based on "The new world atlas of artificial night sky brightness" is the current de facto ALAN standard. It would seem appropriate that the authors compare their work to it since it is currently used by citizen scientists and professionals in a wide range of disciplines. It relates the brightness of the natural night sky to satellite and on the ground SQM measurements. It is a key to dark sky conservation efforts.

The authors use the terms "brightening" and "dimming" repeatedly as undefined terms in the text until the methods section where there is a discussion pointing to equations in Extended Data Table 5. The reader should not have to follow such a twisted path to the understanding of key concepts.

Since the research is based on daily Black Marble NTL product it seems reasonable to expect a concise discussion of it in the text. This discussion should include how the filters used to produce it steer the results.

Fig. 1 : The global ALAN change product is available through the Google Earth Engine application (<https://downloading.projects.earthengine.app/view/alan-change>)

This interface could enable scientists and the enlightened public use this research. Perhaps the authors could consider producing a users guide for this online venue.

Fig. 4: How do the error bars relate to the size of the dots in a1, a2, and a3?

Extended data Fig 4: How do the error bars relate to the size of the dots in a1, a2, and a3 and in c1, c2, and c3?

How can a satellite passing at roughly 1:30 AM characterize changes during the night enabling one to determine "The Night's Rhythm" . SQM observations reveal ALAN changes during the night in cities having a more or less regular pattern which one might call a night's rhythm.

A "novel continuous change detection approach" is mentioned in the Main and "change detection algorithm" is mentioned in Methods. The details seem a bit vaporous. Were sample artificial data sets used as an overall test of the analysis procedure?

This research has an extremely complex layered analysis with many filters and branch points operating on preprocessed data. If that isn't enough the authors state "Reference data for each validation sample was generated through careful visual interpretation by trained analysts using a custom web-based application developed in Google Earth Engine (GEE)." Did these trained analysts observe false negatives and false positives?

Words alone may not be enough to convince the reader as to the validity of the approach. Perhaps graphics relating to intermediate results would help. Did the authors conduct any studies which would give confidence that unique stable results using the start to finish analysis have been achieved?

Are there any ALAN sources the authors could identify and use as an overall check of their methods? Perhaps time series plots of individual pixels for such sources would help to show that an individual pixel is a stable detector.

Version 1:

Reviewer comments:

Referee #1

(Remarks to the Author)

I would like to thank the authors for very thoroughly and successfully revising the text, based on the review comments made.

Referee #2

(Remarks to the Author)

The authors have made a number of significant improvements. The revision of Manuscript #2025-06-15916A moves the paper from very good to excellent. It now does justice to the outstanding research it presents. I recommend it be published.

**Response to Referee #1:**

Response: We appreciate the reviewer for the careful reading and constructive comments. We
addressed each point below and have revised the manuscript accordingly. Please refer to the
**revised clean manuscript** for specific **line numbers** we mentioned in this response.

L116 Since the sinusoidal projection (equal area) is already used by the authors, perhaps it would
be a better option to use it and make the global maps in the paper, as the geographic projection
WGS84 heavily distorts areas and it not a good choice for global maps.

Response: We thank the reviewer for this excellent point. We agree that an equal-area projection
is essential for any quantitative analysis. Accordingly, all our quantitative analyses, including
area calculations and accuracy assessments, were conducted using the 500-m equal-area
sinusoidal projection.

For the global map visualizations in the main text, we have retained the geographic
(latitude/longitude) projection after careful consideration of three key factors:

First, the input of our change detection algorithm -- NASA Black Marble standard products
(VNP46) -- are produced in a geographic longitudinal/latitudinal grid, which is also used for our
original nighttime light change maps. Visualizing the data in this native projection avoids the
need for raster resampling, which inevitably introduces smoothing artifacts or geometric
distortions at the pixel level. Retaining the native grid preserves the visual fidelity of the high-
frequency changes we detected.

Second, the most severe area distortion in the geographic projection occurs at the poles. Since
our study area is strictly limited to the inhabited world between 70°N and 60°S, the extreme
polar distortion is effectively clipped out, maintaining a reasonable visual representation of the
study region. While the sinusoidal projection is mathematically ideal for area calculations (also
needed for sample-based accuracy assessment), it introduces severe shape distortion, which can
hinder the visual interpretation of global patterns.

Third, this approach aligns with standard practices in recent high-profile global remote sensing
studies (e.g., Hansen et al., Science, 2013; Pekel et al., Nature, 2016; Forzieri et al., Nature,
2022), where geographic projection is commonly used for accessibility and interoperability with
standard web-mapping platforms.

We have now added a sentence in the Methods section to make this important distinction
between our analysis and visualization projections clear. The revised text in the manuscript now
reads (Lines 667-672):

"The ALAN change detection results were generated under the linear latitude/longitude
geographic projection, consistent with the input atmospheric- and lunar-BRDF-corrected Black
Marble data. This projection was also retained for visualizations of the global maps. However, all

quantitative analyses, such as area estimates and accuracy assessments, in this study were
conducted following the MODIS/VIIRS 500-meter sinusoidal equal-area projection to ensure
accurate area-based calculations."

While we prefer this projection choice for the reasons outlined above, we fully defer to the
editor's direction and journal house style regarding map projections.

References:

Hansen, M. C., Potapov, P. V., Moore, R., Hancher, M., Turubanova, S. A., Tyukavina, A., ... &
Townshend, J. R. (2013). High-resolution global maps of 21st-century forest cover change.
*science*, 342(6160), 850-853.

Pekel, J. F., Cottam, A., Gorelick, N., & Belward, A. S. (2016). High-resolution mapping of
global surface water and its long-term changes. *Nature*, 540(7633), 418-422.

Forzieri, G., Dakos, V., McDowell, N. G., Ramdane, A., & Cescatti, A. (2022). Emerging signals
of declining forest resilience under climate change. *Nature*, 608(7923), 534-539.

Figure 5.d, in the x-axes, the second 'Jan2020' should perhaps be Jan2021?

Response: We thank the reviewer for pointing out this typo. It has been corrected in the revised
manuscript.

L860 k-hat statistic has also to be reported (not just overall accuracy) given that the two classes
are very different in sample size.

Response: We appreciate the reviewer raising this critical point regarding class imbalance. We
fully agree that in a global study where "change" pixels are rare compared to "stable" pixels,
reporting only Overall Accuracy would be misleading. It is essential to ensure that the metrics
robustly reflect the performance of the minority classes.

Regarding the k-hat (Kappa) statistic, while this has historically been a standard metric in remote
sensing, recent methodological guidelines (e.g., Pontius & Millones, 2011; Olofsson et al., 2014;
Foody, 2020) have shifted away from its use. The consensus in the literature is that Kappa can be
difficult to interpret because it conflates different types of error and is sensitive to class
prevalence in ways that do not necessarily reflect true map accuracy.

To directly address the class imbalance issue that the reviewer highlights, we instead employed
the stratified area-weighted estimator protocol established by Olofsson et al. (2014). This method
is specifically designed to correct for the disparity between sample counts and actual land area.
By weighting the error matrix based on the map area of each stratum, we generate unbiased
User's and Producer's Accuracies that are independent of the class imbalance. We believe these
metrics provide the most transparent and statistically robust view of how well the algorithm
detects the rare "change" events.

To ensure this methodological choice is clear to readers, we have expanded the explanation in the
Methods section to re-emphasize that our protocol was chosen specifically to provide unbiased
estimates in the context of highly imbalanced class. The revised text in the manuscript now reads
(Lines 798-801):

"Accuracy assessment and unbiased area estimates followed established protocols for land
change mapping^{72,73}. An independent validation dataset was generated using a stratified random
sampling design to ensure statistically robust estimations of accuracy by appropriately weighting
the estimators to account for the large differences in mapped area across classes. "

References:

Pontius Jr, R. G., & Millones, M. (2011). Death to Kappa: birth of quantity disagreement and
allocation disagreement for accuracy assessment. *International journal of remote sensing*, 32(15),
4407-4429.

Foody, G. M. (2020). Explaining the unsuitability of the kappa coefficient in the assessment and
comparison of the accuracy of thematic maps obtained by image classification. *Remote sensing
of environment*, 239, 111630.

[72]. Olofsson, P. et al. Good practices for estimating area and assessing accuracy of land
change. *Remote Sens Environ* 148, 42–57 (2014).

[73]. Stehman, S. V., Olofsson, P., Woodcock, C. E., Herold, M. & Friedl, M. A. A global land-
cover validation data set, II: Augmenting a stratified sampling design to estimate accuracy by
region and land-cover class. *Int J Remote Sens* 33, 6975–6993 (2012).

While the strong point of this paper is the use of high temporal resolution (daily) NTL and the
capacity to capture light fluctuations in short time intervals, I believe that the manuscript will
greatly gain by explaining more what is the actual added value of monitoring high temporal
frequency fluctuations globally, with daily data. What exactly do coexisting brightening and
dimming trends show for an area? What is the benefit to know that two regions both exhibit
overall increasing NTL trends but one of the two shows more temporal fluctuation than the
other? It is stated in the text but I think a more focused discussion will improve the objective.

Response: We thank the reviewer for this insightful comment, which highlights the central
contribution of our work. We agree completely that the added value of daily data over monthly
aggregates deserves a more focused explanation.

Our daily analysis provides three key advantages that are lost in monthly or annual composites:

First, precise event attribution. Daily data allows us to pinpoint the exact timing of changes,
enabling direct and unambiguous links between nighttime light dynamics and specific real-world
events. For example, while monthly data might show a general dimming in a region during a
conflict, our daily data can reveal sudden blackouts corresponding to specific military actions or
infrastructure damage on daily basis (e.g., Fig. 5e, Ukraine). This precision transforms NTL

analysis from a tool for observing broad correlations to one capable of near-real-time event
monitoring and impact assessment.

Second, revealing true societal dynamics and volatility. Monthly composites average out
coexisting brightening and dimming trends, masking the true dynamism of a region. To answer
the reviewer's question: an area with high fluctuation (volatility) signifies instability, even if its
net trend is stable. This could reflect energy insecurity, economic precarity, or rapid, disruptive
transformation (e.g., armed conflicts in Syria). In contrast, a region with a similar net trend but
low fluctuation, such as Western Europe, suggests steady, uninterrupted decline. The monthly
data would erroneously show these two vastly different scenarios as being similar. Our daily
approach is the first to globally map this volatility, revealing the complex, simultaneous
processes of development and decline that shape our world.

Third, accurate intensity quantification. Temporal averaging in monthly products inherently
smooths over short-term events, leading to a significant underestimation of their true magnitude.
For instance, a widespread power outage following a hurricane that occurred at the end of one
124 month appears as only a minor dip on the monthly average for that month. Our daily data
captures the full depth and duration of such events (e.g. Fig 1h, Puerto Rico), enabling a more
accurate quantification of their impact on infrastructure, energy systems, and human well-being.

To address this, we have revised the manuscript (see Lines 367-376) to explain why capturing
these high-frequency dynamics is a critical advance as follows:

"Daily satellite NTL observations offer an unprecedented window into societal dynamics,
capturing short-term events typically erased by coarser temporal composites (Fig. 5). This high-
frequency perspective allows us to monitor the "night's dynamics" – the rapid and often abrupt
fluctuations of human activity and societal responses as reflected in ALAN. While annual and
monthly data track broad long-term trends in energy use or ecological impact⁵, our daily-derived
ALAN changes reveal crucial short-term, event-driven dynamics that shape those net long-term
outcomes. By resolving the precise timing of these changes, we can move beyond broad
correlations to link shifts in illumination directly to specific real-world shocks (e.g., military
actions, lockdowns, or policy implementations), while capturing the full magnitude of transient
events (e.g., Fig 1h, Puerto Rico) that would otherwise be smoothed out and underestimated in
monthly averages.

This granular analysis exposes true societal dynamics and volatility that are otherwise masked by
net trends. A central insight from this study is that regions with comparable net annual outcomes
can behave very differently at finer timescales. For instance, an area exhibiting high fluctuation
often signals instability (e.g., armed conflicts in Syria), reflecting energy insecurity, economic
precarity, or sporadic conflict, even if its net trend appears stable. Conversely, a region with low
fluctuation suggests steady, uninterrupted development or decline (e.g., Western Europe).
Distinguishing between these scenarios is critical, as the "night's dynamics" act as a societal
"electrocardiogram," differentiating between resilient systems and those under acute stress."

References:

[5] Elvidge, C. D., Baugh, K., Zhizhin, M., Hsu, F. C. & Ghosh, T. VIIRS night-time lights. in
Remote Sensing of Night-time Light 6–25 (Routledge, 2021).

Two main points to consider:

1) L35, L59-61, “the prevailing assumption of near-universal growth in light radiance” I don’t
think we can say that there is such a unanimous assumption in the literature. Elvidge et al. for
example, identify 68 countries with declining, stable, erratic lights (i.e. not growing). This does
not sound very different compared to the findings in L167-168. Please consider rephrasing.

- Elvidge C et al., 2011, National Trends in Satellite Observed Lighting: 1992-2009, AGU Fall
Meeting Abstracts, <https://ui.adsabs.harvard.edu/abs/2011AGUFMGC32C..03E>

Response: We thank the reviewer for this important point. We agree that our original phrasing
"unanimous assumption" overstated the case, and we appreciate the reference to the important
work by Elvidge et al., which indeed documented country-level declining and stable trends. Our
intention was to challenge the dominant narrative that has focused on net changes at coarse
temporal scales, not to suggest that dimming had never been observed.

To address this, we have revised the text to be more precise and to better distinguish our
findings. The revised sentence in the manuscript now reads (Lines 35-36):

"Our findings challenge the prevailing perspective that changes in light radiance are largely
gradual and unidirectional."

We believe this phrasing is more accurate. It acknowledges that trends other than growth have
been identified before, while highlighting that our key contribution is the discovery, enabled by
daily data, that bidirectional dynamics (coexisting brightening and dimming) are not just
country-level exceptions but a pervasive, high-frequency feature of the global nightscape.

2) L79-80, L158, L355, L386-7, L782-785 etc “Focusing only on the net change year-on-year,
we miss the crucial temporal ...variation..”. In several instances, throughout the text, it is almost
implied that NTL temporal resolution moved from annual to daily, in one step. But in fact, the
temporal NTL resolution was initially annual (DMSP/OLS), then it became monthly (VIIRS
available since 2012) and more recently it became daily (black marble). The monthly data needs
to receive more attention in the manuscript, because it is already available for several years, and
solve the seasonality problem for many applications that do not need the daily resolution.
Monthly data is a good basis for most analysis within the year, accounting for phenology etc.
And several such applications have appeared using monthly data to examine virus spread,
population seasonality etc. The clear acknowledgment of the monthly VIIRS data will also help

the authors to better focus on the benefit of the daily data. What can daily data do that monthly
can not? Also note, that monthly averages are in fact used in some parts of the manuscript (L391,
L637-642). Again the benefit of using daily data (instead or monthly) should become clearer.

[https://developers.google.com/earth-](https://developers.google.com/earth-engine/datasets/catalog/NOAA_VIIRS_DNB_MONTHLY_V1_VCMSLCFG)
[engine/datasets/catalog/NOAA_VIIRS_DNB_MONTHLY_V1_VCMSLCFG](https://developers.google.com/earth-engine/datasets/catalog/NOAA_VIIRS_DNB_MONTHLY_V1_VCMSLCFG)

Response: We thank the reviewer for this excellent and critical point. We agree completely that
the manuscript needs to acknowledge the progression of NTL data temporal resolutions,
particularly the use of monthly data, and discuss the unique scientific value of daily data
compared to the established monthly products.

To address this, we have made two key revisions. First, we have revised the introduction to
properly contextualize the evolution of NTL data from annual (DMSP-OLS and VIIRS) to the
valuable monthly with more citations added, and now daily (VIIRS Black Marble) products. The
revised sentence in the manuscript now reads (Lines 69-79):

"This understanding has largely been shaped in part by the broad range of nighttime light (NTL)
data at different temporal resolutions^{5,6}, collected by satellite sensors that measure the artificial
lights' radiance using visible and near-infrared light reflected or emitted at night, enabling
consistent monitoring of human settlements and activities from space. Annual and multi-year
composites, such as those produced from earlier Defense Meteorological Satellite Program's
Operational Linescan System (DMSP-OLS)⁷ and later from Visible Infrared Imaging Radiometer
Suite (VIIRS) Day/Night Band (DNB) data^{8,9}, have been instrumental in mapping long-term
ALAN trends^{10,11}, documenting the expansion of urban extant¹², and estimating the light
pollution effects^{13,14}. Monthly VIIRS DNB products balance noise reduction with improved
temporal granularity, enabling analyses of intra-annual variability^{1,15}, such as seasonal impacts¹⁶
and epidemiological dynamics^{17,18}."

Second, and more central to your comment, we have added new paragraphs in the section
"Capturing the Night's Dynamics with Daily Data" to explain the specific advantages of using
daily data for change detection. The revised sentence in the manuscript now reads (Lines 95-99):

"More recently, advances in NASA's daily Atmospheric- and Lunar-BRDF-corrected Black
Marble NTL product²³ have provided an unprecedented opportunity to quantitatively
characterize Earth's finer daily nighttime dynamics. Its transformative value is not only in its
daily temporal resolution, complements the short-term and abrupt fluctuations that are otherwise
averaged out in coarser composites, but also in the scientific rigor of its design."

References:

[1] Levin, N. et al. Remote sensing of night lights: A review and an outlook for the future.
Remote Sens Environ 237, (2020).

[5] Elvidge, C. D., Baugh, K., Zhizhin, M., Hsu, F. C. & Ghosh, T. VIIRS night-time lights. in
 Remote Sensing of Night-time Light 6–25 (Routledge, 2021).

[6] Zheng, Q., Weng, Q., Zhou, Y. & Dong, B. Impact of temporal compositing on nighttime
 light data and its applications. *Remote Sens Environ* 274, 113016 (2022).

[7] Elvidge, C. D., Baugh, K. E., Kihn, E. A., Kroehl, H. W. & Davis, E. R. MAPPING CITY
 LIGHTS WITH NIGHTTIME DATA FROM THE DMSP OPERATIONAL LINESCAN
 SYSTEM. *Photogramm Eng Remote Sensing* 63, (1997).

[8] Elvidge, C. D., Zhizhin, M., Ghosh, T., Hsu, F. C. & Taneja, J. Annual Time Series of Global
 VIIRS Nighttime Lights Derived from Monthly Averages: 2012 to 2019. *Remote Sens (Basel)*
 13, 922 (2021).

[9] Wang, Z., Shrestha, R. M., Roman, M. O. & Kalb, V. L. NASA’s Black Marble Multiangle
 Nighttime Lights Temporal Composites. *IEEE Geoscience and Remote Sensing Letters* 19,
 (2022).

[10] Kyba, C. C. M. et al. Artificially lit surface of Earth at night increasing in radiance and
 extent. *Sci Adv* 3, e1701528 (2017).

[11] Hu, Y. & Zhang, Y. Global nighttime light change from 1992 to 2017: Brighter and more
 uniform. *Sustainability* 12, 4905 (2020).

[12] Zhou, Y., Li, X., Asrar, G. R., Smith, S. J. & Imhoff, M. A global record of annual urban
 dynamics (1992–2013) from nighttime lights. *Remote Sens Environ* 219, 206–220 (2018).

[13] Falchi, F. et al. The new world atlas of artificial night sky brightness. *Sci Adv* 2, e1600377
 (2016).

[14] Kyba, C. C. M., Altıntaş, Y. Ö., Walker, C. E. & Newhouse, M. Citizen scientists report
 global rapid reductions in the visibility of stars from 2011 to 2022. *Science* (1979) 379, 265–268
 (2023).

[15] Baugh, K., Hsu, F.-C., Elvidge, C. D. & Zhizhin, M. Nighttime lights compositing using the
 VIIRS day-night band: Preliminary results. *Proceedings of the Asia-Pacific Advanced Network*
 35, 70–86 (2013).

[16] Levin, N. The impact of seasonal changes on observed nighttime brightness from 2014 to
 2015 monthly VIIRS DNB composites. *Remote Sens Environ* 193, 150–164 (2017).

[17] Yoneoka, D. et al. Indirect and direct effects of nighttime light on COVID-19 mortality
 using satellite image mapping approach. *Sci Rep* 14, 25063 (2024).

[18] Bharti, N. et al. Explaining seasonal fluctuations of measles in Niger using nighttime lights
 imagery. *Science* (1979) 334, 1424–1427 (2011).

[23] Román, M. O. et al. NASA’s Black Marble nighttime lights product suite. *Remote Sens*
 *Environ* 210, 113–143 (2018).

Some minor points:

3) L81-82 “This oversight [using only annual data] limits our ability to truly link cause and
effect...” Similarly to spatial resolution, the highest technically available temporal resolution is
not by default best for any application. Please consider better explaining this sentence.

Response: We thank the reviewer for this crucial insight. We fully agree that the highest temporal
resolution is not automatically the best for every application. As the reviewer notes, daily data
contains significant noise, and for monitoring slow, steady processes like long-term urbanization,
temporally aggregated products (monthly or annual) are often superior as they effectively
suppress this noise.

However, our study specifically aims to attribute ALAN changes to discrete, short-term drivers,
such as natural disasters, conflicts, or sudden policy shifts. In this specific context, the "temporal
smoothing" inherent in composites becomes a limitation rather than a benefit. For example, a
severe, short-term power outage integrated into a monthly average appears only as a slight dip in
radiance, making it statistically indistinguishable from normal fluctuation. To "truly link cause
and effect" for these events, we require the daily precision to identify the exact onset of the
change, despite the challenge of higher noise levels.

To address the reviewer's concern, we have revised the manuscript to explicitly acknowledge the
trade-off: while aggregation reduces noise, high-frequency data is required to resolve the
"temporal mismatch" between rapid real-world events and satellite observations. The revised text
now reads (Lines 77-81):

"Monthly VIIRS DNB products balance noise reduction with improved temporal granularity,
enabling analyses of intra-annual variability^{1,15}, such as seasonal impacts¹⁶ and epidemiological
dynamics^{17,18}. However, despite these strengths, temporal aggregation inherently masked short-
term fluctuations and the bidirectional changes arising from the coexistence of intra-annual
brightening and dimming."

4) L93 It would be useful to briefly summarize sources and magnitude of noise here.

We thank the reviewer for this excellent suggestion. We have revised the text to better summarize
these factors, the revised text now reads (Lines 106-111):

"However, this abundance of daily data comes with its analytical hurdles. The signal is subject to
substantial noise from a variety of sources, including atmospheric interference, variations in
sensor viewing and local geometry, and ephemeral events like snow cover²⁵. The magnitude of
this combined noise is highly variable and can often exceed the subtle, real-world changes we
aim to detect, making most current algorithms ineffective at separating the true signal from the
noise."

This revision now explicitly lists the primary sources of noise. Regarding the magnitude, we
have clarified that it is substantial and highly variable, so much so that a single quantitative value

would be misleading. We have also cited Wang et al., (2021), if the reader wants to know more
details. We emphasize that the noise can often overwhelm the signal, which provides critical
context for why our sophisticated change detection algorithm is necessary.

Reference:

[25]Wang, Z., Román, M. O., Kalb, V. L., Miller, S. D., Zhang, J., & Shrestha, R. M. (2021).
Quantifying uncertainties in nighttime light retrievals from Suomi-NPP and NOAA-20 VIIRS
Day/Night Band data. *Remote Sensing of Environment*, 263, 112557.

5) L169 Please check author names.

Response: We thank the reviewer for pointing out this typo, it has been corrected in the revised
manuscript.

6) L180 Please check “oremerging”

Response: We thank the reviewer for pointing out this typo, it has been corrected in the revised
manuscript.

7) L225 According to EUROSTAT the population at the EC 27 level (The European Union’s 27
member states) is not declining. The population in France, the UK and in the Netherlands is also
not declining. In fact, it is slightly increasing

(https://ec.europa.eu/eurostat/databrowser/view/tps00001/default/table?lang=en&category=t_de
[mo.t demo pop](https://ec.europa.eu/eurostat/databrowser/view/tps00001/default/table?lang=en&category=t_de))

Response: We thank the reviewer for this important correction. You are correct that our original
statement was inaccurate. We have now removed this statement in our revised manuscript.

8) L226 Please consider moving Venezuela to an other paragraph, as the discussion here is for
Europe.

Response: Agree and we have moved the discussion of Venezuela to a different paragraph. The
revised text now reads (Lines 244-249):

9) L268-283 Could the discussion become more specific to better show the causes of NTL
fluctuation? How urban density and evolving land use patterns affect NTL trends?

Response: We thank the reviewer for this excellent suggestion. We have substantially revised the
"Change Intensity Versus Change Area" section (Lines 275-313) to discuss the causes of NTL
fluctuation and how urban density and land use affect NTL trends. The revised text now reads
(Lines 296-310):

"This pattern is closely tied to the country's dominant land-use trajectory: low-density suburban
sprawl^{52,53}. Unlike high-density redevelopment, horizontal expansion is additive, distributing
artificial light across vast peri-urban areas without requiring the disruption of existing
infrastructure, resulting in widespread but gradual, low-volatility trends^{54,55}. India presents a
similar profile of widespread area change accompanied by moderate intensity changes (8% of
initial radiance for brightening, -9% of initial radiance for dimming), contrasting with China,
where more intense transformations prevail (12% of initial radiance for brightening, -14% of
initial radiance for dimming). China's high-intensity signals are consistent with a strategy of
vertical urbanization and high-density land conversion, where single-story structures or
agricultural lands are rapidly demolished and replaced by concentrated high-rise complexes^{56,57}.
This specific evolution of land use creates a unique NTL signature: distinct periods of dimming
(demolition) followed by explosive brightening (vertical reconstruction). Consequently, higher
urban density acts as a multiplier for NTL volatility, whereas lower-density sprawl acts as a
dampener, capturing the structural evolution of the built environment itself."

References:

[52] Wang, M. & Debbage, N. Urban morphology and traffic congestion: Longitudinal evidence
from US cities. *Comput Environ Urban Syst* 89, 101676 (2021).

[53] Hamidi, S. & Ewing, R. A longitudinal study of changes in urban sprawl between 2000 and
2010 in the United States. *Landsc Urban Plan* 128, 72–82 (2014).

[54] Wu, Y., Li, C., Shi, K., Liu, S. & Chang, Z. Exploring the effect of urban sprawl on carbon
dioxide emissions: An urban sprawl model analysis from remotely sensed nighttime light data.
*Environ Impact Assess Rev* 93, 106731 (2022).

[55] Frantz, D. et al. Unveiling patterns in human dominated landscapes through mapping the
mass of US built structures. *Nat Commun* 14, 8014 (2023).

[56] Mahtta, R., Mahendra, A. & Seto, K. C. Building up or spreading out? Typologies of urban
growth across 478 cities of 1 million+. *Environmental Research Letters* 14, 124077 (2019).

[57] Zhou, L., Dang, X., Mu, H., Wang, B. & Wang, S. Cities are going uphill: Slope gradient
analysis of urban expansion and its driving factors in China. *Science of The Total Environment*
775, 145836 (2021).

10) Figure 4.b it is a bit surprising to see e.g. Italy and France in the same class with Syria and
Yemen, all considered high baseline nations, now experiencing dimming. I think that the
dimming in these countries is attributed to very different reasons (policy/technology
advancements vs extensive conflicts). See also my next comment below about the typology.

Response: We thank the reviewer for this very insightful observation. The reviewer is correct: the
underlying drivers for dimming in Europe are fundamentally different from those in Syria and
Yemen. Our typology in the original Fig. 4b was intentionally a high-level classification based
on just two quantitative metrics: a country's baseline ALAN in 2014 (high vs. low) and its net
radiance trend over the study period (brightening vs. dimming). By these strict criteria, France

and Syria are grouped together in the "Post-Peak and Waning" category as they both started with
high baseline (above median) lighting and experienced a net decrease. Because this typology was
based solely on two ALAN radiance-related criteria, its initial baseline level and its trend, the
resulting categories sometimes obscured important contextual differences.

Considering this Fig. 4b has contributed limited information and the original Fig. 4c already
provides information of increasing volatility in nighttime light (which the original Fig. 4c already
provides), we have decided to remove this Fig. 4b and focused on Fig. 4c (now Fig. 4b) to
resolve the confusion and save space for the key findings. We believe this change improves
clarity and prevents over-generalization while retaining the core results relevant to the
manuscript's main findings.

11) L880 The typology contains the class "(de)-electrification" for stable regions, with both
growing or declining trends. An interesting finding of the paper is that some developed counties
show declining trends, attributed to a policy and/or technological shift, adopting LED tech etc
(L220-221, L224, L233-4, L417-421). Therefore, in my opinion, it would be interesting to split
this class in two, and have a separate class as stable+declining. The new class would be
interesting as it is a totally different case, compared to de-electrification due to crisis and
disasters. More countries will probably be added to this class in the future. Also it supports the
point made in the manuscript for a need to rethink the association of NTL with socioeconomic
variables (L417). This is the main point to rethink in my view. A future negative correlation of
development with NTL. France in Figure 5.c is a clear example.

Response: We thank the reviewer for this excellent suggestion to refine our driver's typology. We
agree that the processes behind "electrification" (increasing radiance) and "de-electrification"
(decreasing radiance) are fundamentally different. As suggested, we have now split the original
'(de)-Electrification' class into two separate categories in our driver typology (Extended Data
Table 4) and all related analyses: (i). Electrification: Occurs in regions stable land cover and
population but substantial growth in electric infrastructure (Brightening). (ii). De-electrification:
Occurs in regions with stable land cover and population but a substantial decline in ALAN
(Dimming), often associated with energy-saving measures like LED streetlight renovation or
declines in energy access. We have also updated Fig. 2 and Extended Data Fig. 1 to reflect this
more precise classification.

12) L870 the % change would also be useful. Possibly by removing the radiance column that is
not further used to show change, to make space in the table.

Response: We thank the reviewer for this thoughtful suggestion. We have revised the Extended
Data Table 2 accordingly.

L421 the use of this reference needs a bit more explaining [46]. Is it the correct reference

Response: We thank the reviewer for this important query. You are correct to question this
reference in its original context; its placement was misleading. The reference (Plagborg-Møller
& Wolf, 2021) is an important paper in econometrics regarding the analysis of dynamic systems
and their response to "shocks." We cited it to provide theoretical support for our central
argument: that simple, static correlations are insufficient for understanding a system driven by
abrupt events. Events like strong policy interventions or rapid technological transitions act as
exogenous shocks to the socio-technical system producing ALAN. The system's response is
therefore dynamic and path-dependent, and analyze these relationships requires comprehensive
modeling of multivariate time series to understand the impulse response functions of endogenous
variables. To make this link explicit and precise, we have revised the text and added a classic
reference (Stock & Watson, 2001) to ground the argument. The revised text now reads (Lines
446-451):

"Such events act as exogenous shocks to the socio-technical system, creating dynamic, time-
lagged interdependencies that require robust empirical modeling⁶³. One of the approaches that is
a good fit for this involves the simultaneous analysis of multivariate time series. Advanced
methodologies from econometrics, specifically those analyzing impulse response functions in
multivariate time series, offer the necessary tools to disentangle these complex interactions⁶⁴."

We believe this revision clearly explains the rationale for referencing econometric methodologies
in the context of ALAN dynamics.

References:

[63] Plagborg - Møller, M. & Wolf, C. K. Local projections and VARs estimate the same
impulse responses. *Econometrica* 89, 955 – 980 (2021).

**Referee #2 (Remarks to the Author):**

Response: We appreciate the reviewer for the careful reading and constructive comments. We
addressed each point below and have revised the manuscript accordingly. Please refer to the
**revised clean manuscript** for **specific line numbers** we mentioned in this response.

Given the definition of artificial, it is essential that ALAN be measured relative to or at least
thoughtfully compared to the brightness of the natural night sky. The intensity of ALAN ranges
from a maximum, usually in large cities, to a value of near zero in remote locations where natural
night sky brightness variations dominate. The authors designation of “Persistently dark regions”
are presumably such locations. These areas of natural night sky brightness should be included in
the analysis. Typically the brightness of the natural airglow can vary by 40% or more even
during the course of a single more or less normal night. In areas of slight to moderate ALAN,
variations in the brightness of the natural night sky print through. Failure to account for this
situation will result in erroneous interpretations of the data.

Response: We thank the reviewer for raising this point about the potential influence of natural
night sky brightness on measurements of ALAN emissions from Earth’s surface. This
manuscript’s primary focus is artificial lighting changes, not airglow or skyglow. While airglow
is present (and visible) in the VIIRS DNB data, it is normally around $\sim 0.15\text{-}0.2$ nW/cm²sr,
which is below the minimum radiance change threshold used in our analysis (1.0 nW/cm²sr).

Natural airglow is present across the entire globe, including 'persistently dark regions', where the
signal is dominated by this natural background rather than anthropogenic emissions, and
including these regions in the analysis would introduce significant noise and dilute the study’s
focus on direct emissions of artificial light. However, we agree that it is important to account for
the possibility of variability in natural airglow effects on our observations of artificial emissions.
Our change detection methodology was specifically designed to be robust to these natural
variations in several ways as follows:

First, our time series-based approach inherently adapts to the local baseline brightness. It does
not assume an absolute zero-light background, instead, it learns the typical radiance level at each
pixel which accounted for the contribution from natural background. Changes are detected as
significant deviations from this established local baseline (including average airglow and its
seasonal variations), not from an absolute zero radiance. This ensures that we are measuring
changes in artificial light relative to the ambient conditions.

Second, to further address the variability of airglow and other variable factors, our algorithm
includes two strict criteria that filter out transient, low-magnitude fluctuations. First, we only
consider changes that exceed a radiance threshold of 1.0 nW \cdot cm⁻² \cdot sr⁻¹. Even if natural
airglow varies by 40% (e.g., from 0.2 to 0.28 nW), this fluctuation remains far below our
detection threshold. This conservative cutoff effectively prevents natural variations from

registering as artificial changes. Second, our primary defense against the large variability of
high-frequency data (e.g., noise from airglow or aurora; see an example in Fig. R1) is require
change to be confirmed in consecutive observations. While natural light can spike significantly
on a single night, it rarely exhibits the sustained, step-like behavior of human infrastructure
changes.

We have revised our Methods section to make it clearer to the reader. The revised text now reads
(Lines 723-726):

"This ensures that only sustained and statistically robust deviations from the local baseline are
identified. This design is well-suited to capturing sudden human-driven ALAN change events,
which typically manifest as persistent and significant radiance deviations."

**Fig. R1.** Example of a high-latitude area with aurora activity, showing no artificial nighttime
light change. Very high-resolution images from Google Earth are provided to provide context.
VZA-COLD labels this as stable (no change).

The authors present specific examples that ALAN provides a unique view of many aspects of
human activity making their results potentially useful to a wide variety of research fields. If this
article is being submitted to Nature it should be written to communicate with researchers in a
wide variety of fields. Words alone may not be sufficient. Several flow chart/organization types
of graphics could help the serious reader to visualize and utilize the concepts and procedures
described in this paper. Given the resolution of the figures in the manuscript, the authors should
consider adding an extensive online interactive component so readers can ask their own
questions and see how the data and analysis supports the conclusions presented in the text.

**Response:** We thank the reviewer for these excellent suggestions for improving the accessibility
and utility of our research. We have taken several steps to make our methods and data more
transparent and interactive:

(i) As suggested, we have added a comprehensive flowchart of our entire analytical workflow in
Extended Data Fig. 1. This graphic provides a clear, step-by-step visual guide to our procedures.

(ii) We have developed a Google Earth Engine (GEE) application that allows for interactive
exploration of our full-resolution, pixel-based ALAN change maps. This interface enables users
to directly visualize the data, query specific locations, and see how our analysis supports the
paper's conclusions.

(iii) Our ALAN change products are open access for peer review purposes at this link
(https://drive.google.com/file/d/19jyrh4loD-szGZRv5S6RMSNf_ZpHNwq/view?usp=sharing).
Following publication of our manuscript, we will immediately release the products for public in
our GitHub repository (<https://github.com/GERSL/VZA-COLD>), along with technical
specifications. Readers and users can also make comments or ask questions online by creating
issues here <https://github.com/GERSL/VZA-COLD/issues>.

In addition, our ALAN change product has been selected as an official NASA product. It will be
publicly accessible via the NASA data portal as part of the Black Marble NTL product suite in
2026 (expected as VNP46A5). This integration ensures broad and long-term availability of the
dataset to the scientific community and other stakeholders, enabling interactive exploration,
visualization, and analysis of high-frequency ALAN dynamics on a global scale. By embedding
the product within the NASA ecosystem, users will be able to leverage standardized tools,
documentation, and APIs for seamless integration into their research workflows.

An interactive light pollution map based on “The new world atlas of artificial night sky
brightness” is the current de facto ALAN standard. It would seem appropriate that the authors
compare their work to it since it is currently used by citizen scientists and professionals in a wide
range of disciplines. It relates the brightness of the natural night sky to satellite and on the
ground SQM measurements. It is a key to dark sky conservation efforts.

Response: We thank the reviewer for this important comment and for highlighting the New
World Atlas of Artificial Night Sky Brightness. We fully agree that the Atlas is the standard for
assessing the state of artificial sky brightness and is vital for conservation efforts. However,
based on this comment, we realize that our original manuscript may not have been sufficiently
clear regarding the fundamental physical differences between skyglow and direct light emissions
from the Earth’s surface.

This study focuses on direct artificial light emissions (upward radiance measured by satellites),
not skyglow (downward scattered light reported by the Atlas or measured by ground sensors).
While related, these are distinct physical phenomena. The Atlas/SQM values refer to the radiance
of light scattered by the atmosphere back to the ground. This integrates light from sources over
large areas (~10 km). Our Study measures the source dynamics -- the light emitted directly from
specific pixels (~750 m) into space.

A direct quantitative comparison between our dynamic, daily change detection and the World
Atlas is not methodologically valid without complex radiative transfer modeling, which is
outside the scope of this work. For example, a localized change in streetlighting shielding might
significantly reduce the lights contribution to skyglow, without altering the upward radiance
detected by VIIRS from light reflected from the street surface. Therefore, our work is not a
replacement for the Atlas, but a complement that reveals the high-frequency source dynamics
driving the impacts modeled in products like the Atlas.

To address the potential confusion, we have revised the Discussion to explicitly state that current
satellite NTL observations capture direct changes in upward light, whereas ground-based
monitoring captures the integrated scattering effect from downward skyglow. These represent
distinct physical quantities that are not directly comparable but provide complementary
information regarding the night environment. These represent distinct physical quantities that are
not directly comparable but provide complementary information regarding the night
environment. The revised text now reads (Lines 461-468):

"However, it is crucial to recognize that current operational satellite NTL observations, such as
those from VIIRS DNB, primarily capture post-midnight light emitted upward from the surface,
whereas ground-based monitors like those used for the World Atlas measure downward
skyglow¹³. These represent distinct physical quantities that are not directly comparable but
provide complementary information regarding the night environment. Additionally, VIIRS DNB
is most sensitive to a specific spectral range (approximately 500-900 nm), representing a societal
blind spot for trends occurring earlier in the evening or in different spectral bands^{1,68}."

References:

[1] Levin, N. et al. Remote sensing of night lights: A review and an outlook for the future.

Remote Sens Environ 237, (2020).

[13] Falchi, F. et al. The new world atlas of artificial night sky brightness. Sci Adv 2, e1600377
(2016).

[68] Falchi, F. et al. The new world atlas of artificial night sky brightness. Sci Adv 2, e1600377
(2025).

The authors use the terms “brightening” and “dimming” repeatedly as undefined terms in the text
until the methods section where there is a discussion pointing to equations in Extended Data
Table 5. The reader should not have to follow such a twisted path to the understanding of key
concepts.

Response: We thank the reviewer for this crucial feedback. We completely agree that the core
concepts of "brightening" and "dimming" should be clearly defined for the reader at their first
introduction, rather than much later in the Methods. To correct this, we have added a concise,

conceptual definition of these key terms early in the Introduction. The revised text now reads
(Lines 60-66):

"In this study, we define brightening changes as sustained increases in nighttime radiance
resulting from ALAN transitions, which may stem from abrupt events like new construction or
lighting installations, or gradual trends such as urban densification and urban expansion.
Conversely, dimming changes are defined as sustained decreases in ALAN radiance, caused by
abrupt disruptions such as power outages or armed conflicts, or longer-term processes including
depopulation, economic decline, or energy-saving interventions (illustrated in Extended Data
Fig. 5)."

Since the research is based on daily Black Marble NTL product it seems reasonable to expect a
concise discussion of it in the text. This discussion should include how the filters used to produce
it steer the results.

Response: We thank the reviewer for this excellent point. We agree that a concise description of
the Black Marble product, and particularly the key corrections that enable our analysis, is
essential context for the reader.

The reviewer is correct that the specific processing steps, or "filters", that NASA applies to create
Black Marble are what "steer the results." By rigorously correcting for Atmospheric interference
(aerosols, water vapor) and Lunar BRDF effects (moonlight reflection anisotropy), these filters
"steer" the analysis away from environmental noise and toward the true anthropogenic signal.
Without them, the raw daily NTL signal would be effected by lunar phases and atmospheric
variability, making the detection of subtle ground changes impossible. To address this, We
revised this paragraph to provide more information of the black marble products and added a
sentence to explain the filtered observations (e.g., snow and clouds). The revised text now reads
(Lines 99-103):

"Unlike earlier nightlight products that primarily served visualization or coarse trend analysis,
Black Marble applies comprehensive corrections for atmospheric conditions, terrain, and lunar
illumination to improve radiometric stability, and at the same time provides information of the
contaminated pixels, such as clouds and snow^{23,24}."

References:

[23] Román, M. O. et al. NASA's Black Marble nighttime lights product suite. Remote Sens
Environ 210, 113–143 (2018).

[24] Frey, R. A., Ackerman, S. A., Holz, R. E., Dutcher, S. & Griffith, Z. The Continuity
MODIS-VIIRS Cloud Mask. Remote Sensing 12, 3334 (2020).

Fig. 1: The global ALAN change product is available through the Google Earth Engine
application (<https://downloading.projects.earthengine.app/view/alan-change>)
This interface could enable scientists and the enlightened public use this research. Perhaps the
authors could consider producing a users guide for this online venue.

Response: We thank the reviewer for this constructive suggestion. We agree that clear
documentation is essential to maximize the utility of this tool for both the scientific community
and the general public. We added a user guide for the Google Earth Engine interface to facilitate
immediate exploration of the data ([https://github.com/GERSL/VZA-COLD/wiki/User-Guide-
for-the-GEE-Application](https://github.com/GERSL/VZA-COLD/wiki/User-Guide-for-the-GEE-Application)), and will continue to update it depending on users' feedback.
Furthermore, as this dataset has been selected as a future official NASA product (VNP46A5), a
comprehensive, official NASA data guide will be developed based on this current version to
support long-term global distribution and application.

Fig. 4: How do the error bars relate to the size of the dots in a1, a2, and a3?

Response: We thank the reviewer for this question. To clarify, the dots in Fig. 4a1, a2, and a3
represent the total annual sum of ALAN radiance change for each year. The size of the dots is
uniform and does not represent a variable. The uncertainty in the figure is shown by the shaded
area, which represents the 95% confidence interval of the Sen's slope trend line, rather than the
measurement error of individual annual data points.

Extended data Fig 4: How do the error bars relate to the size of the dots in a1, a2, and a3 and in
c1, c2, and c3?

Response: We thank the reviewer for this question. To clarify, the dots in Fig. 4a (panels a1, a2,
and a3) represent the calculated global annual sum of ALAN radiance change for each specific
624 year. The size of these dots is uniform and does not represent a variable. The "error bars" or
625 uncertainty visualization refers to the shaded area, which represents the 95% confidence interval
of the estimated Sen's slope trend line, rather than the measurement error of individual annual
data points.

How can a satellite passing at roughly 1:30 AM characterize changes during the night enabling
one to determine "The Night's Rhythm". SQM observations reveal ALAN changes during the
night in cities having a more or less regular pattern which one might call a night's rhythm.

Response: We thank the reviewer for this keen observation. We agree that the term "Night's
Rhythm" could be misinterpreted as referring to diurnal (intra-night) patterns, which indeed
cannot be captured by a sun-synchronous satellite passing at ~1:30 AM. Our intention was to

describe the day-to-day fluctuations of human activity, but we agree that "Dynamics" is a
scientifically more precise term that avoids confusion with nightly cycles. Accordingly, we have
replaced the phrase "Night's Rhythm" with "Night's Dynamics" throughout the manuscript.

A "novel continuous change detection approach" is mentioned in the Main and "change detection
algorithm" is mentioned in Methods. The details seem a bit vaporous. Were sample artificial data
sets used as an overall test of the analysis procedure?

Response: We thank the reviewer for this important feedback regarding the clarity of our
methodology. We agree that the term "novel" was imprecise given that our core approach is a
global-scale adaptation of the verified VZA-COLD algorithm (Li et al., 2022). We have removed
the word "novel" from the manuscript to accurately reflect that this is an expansion and
refinement of an established method.

To address the reviewer's concern that the details were "vaporous," we have also expanded the
Methods section to more clearly outline the validation and testing procedure, and added one
figure to show the detailed workflow of the method (Extended Fig. 5).

Though the foundational VZA-COLD algorithm was extensively tested for sensitivity and
robustness in its original publication, for this study, we made specific refinements to algorithm
parameters to ensure consistent performance across the globe's diverse environmental conditions.
This was done using a dedicated set of calibration samples, which were entirely independent of
our final validation dataset (we added some sentences to explain this in Supplementary
Information). The final ALAN change maps were then independently validated using a stratified
random sample of over 4,000 units globally, as detailed in our Accuracy Assessment section. The
added text now reads (Supplementary Lines 169-173):

"A global calibration dataset of 610 representative pixels, previously compiled from diverse
geographic regions, land cover types, and latitudes for algorithm development¹ was applied to
decide the optimal skipping fraction and moving window length, ensuring robust and consistent
detection of ALAN changes while maintaining computational efficiency."

References:

[1]Li, T., Zhu, Z., Wang, Z., Román, M. O., Kalb, V. L., & Zhao, Y. (2022). Continuous
monitoring of nighttime light changes based on daily NASA's Black Marble product suite.
Remote Sensing of Environment, 282, 113269.

This research has an extremely complex layered analysis with many filters and branch points
operating on preprocessed data. If that isn't enough the authors state "Reference data for each
validation sample was generated through careful visual

interpretation by trained analysts using a custom web-based application developed in Google
Earth Engine (GEE).” Did these trained analysts observe false negatives and false positives?

Response: We appreciate the opportunity to clarify this aspect of our methodology. It appears
that our initial text may not have explicitly detailed how specific error types were distinguished
during the interpretation process. To answer your question directly: yes, our validation protocol
was explicitly designed for analysts to systematically identify and record both commission errors
(false positives) and omission errors (false negatives).

For every validation sample, trained analysts compared the map's output to the full NTL time
series and ancillary data, including high-resolution optical imagery, news, and social media.
They were required to label every instance of a false positive (where the algorithm detected a
change that was not real) and a false negative (where the algorithm missed a real, observable
change). These recorded errors were then directly and quantitatively accounted for in the
confusion matrices (Extended Data Table 1) and used to calculate our unbiased, area-weighted
accuracy estimators. This ensures that our reported accuracy is a true reflection of the algorithm's
performance on real-world data.

Furthermore, regarding the interpretation of causal drivers: To ensure consistency and reliability
for this more complex task, a strict quality control protocol was implemented. Each sample
identified as "change" was independently interpreted by two trained analysts to determine the
driver. If there were any discrepancies in their labels, the sample was flagged and adjudicated by
a third, senior analyst to produce a final, consensus label.

We have added sentences to the Methods section to provide more details on these analysis
procedures. The added text now reads (Lines 821-832):

"During this process, analysts explicitly identified discrepancies between the map and reference
labels, recording specific instances of commission errors (false positives) and omission errors
(false negatives) to ensure robust accuracy estimation. For samples where the mapped change
agrees with the reference data, interpreters also qualitatively noted the potential direct casual
drivers defined in Extended Data Table 4, based on the information from the remote sensing
images and other open-access resources, such as the VIIRS nightfire gas flaring data from
SkyTruth⁷⁴, conflict data from UCDP (Uppsala Conflict Data Program), earthquake data from
CrisisWatch, hurricane records from NOAA (National Oceanic and Atmospheric
Administration), social media (e.g., X and Weibo), and relevant news and report. To ensure
objectivity in this driver attribution, we implemented a strict quality control protocol: each
sample was independently interpreted by two trained analysts. Any discrepancies between their
driver labels were flagged and adjudicated by a third, senior analyst to produce the final
consensus label."

References:
[74]Elvidge, C. D., Zhizhin, M., Hsu, F.-C. & Baugh, K. E. VIIRS nightfire: Satellite pyrometry
at night. Remote Sens 5, 4423–4449 (2013).

Words alone may not be enough to convince the reader as to the validity of the approach.
Perhaps graphics relating to intermediate results would help. Did the authors conduct any studies
which would give confidence that unique stable results using the start to finish analysis have
been achieved?

Response: We thank the reviewer for raising this critical issue. To address this, we have made
two key additions to the manuscript. First, we added the Supplementary Fig. 4 to show the
intermediate results (observations, time series models, break points) against an NTL change
event to further convince the reader the validity of the approach. Second, we did test our change
maps in several unique stable places (e.g., remote high-latitude areas and deserts) to make sure
our change maps did not show large artifact (or commission errors) from the start to finish (see
Fig. R2).

**Fig. R2. Examples of dark regions in the global ALAN change product (2014-2022).** a. The
accumulated ALAN change time map for abrupt changes detected in the example locations. The
colors show the year of the most recently detected abrupt changes. The accumulated change time map

for gradual changes can be found in Fig. 1b. **b-i.** Zoomed-in examples illustrating the accumulated
most recent abrupt and gradual ALAN change time in different locations (locations indicated by
white boxes in **a**). **b.** Amazon rainforest. **c.** Northern Chad and Northern Sudan areas in Sahara
Desert region. **d.** Central Yakutia in Siberian Taiga. **e.** Australian western deserts. **f.** Nunavut tundra
of Northern Canadian Shield. **g.** Greenland. **h.** Congo Basin Rainforest. **i.** Mountainous rainforests
in central Papua and Papua New Guinea. Note that almost no commission errors could be find in
all those places in our gradual and abrupt change maps of all years.

Are there any ALAN sources the authors could identify and use as an overall check of their
methods? Perhaps time series plots of individual pixels for such sources would help to show that
an individual pixel is a stable detector.

Response:

We thank the reviewer for this suggestion. It is important to clarify that our product has already
undergone a rigorous, comprehensive validation process that goes well beyond looking for
individual examples.

First, as detailed in the "Accuracy Assessment" section, we employed a probability-based
stratified random sampling design—the gold standard for remote sensing product validation
(Olofsson et al., 2014). We evaluated over 4,000 independent sample units (2,093 for abrupt
changes and 1,930 for gradual changes) distributed globally. This robust statistical approach
confirms the high accuracy of our product and ensures that our results are not artifacts but true
detections of ALAN change.

Second, the core algorithm (VZA-COLD) has been rigorously verified in our previous work (Li
et al., 2022). To illustrate the pixel-level stability requested by the reviewer, we have included
some figures from that study. Fig. R3 demonstrates how our View Zenith Angle (VZA)
stratification reduces uncertainty in daily data by accounting for angular effects. The new
Supplementary fig. 3b shows the algorithm's ability to detect change that is only visible at
specific viewing angles (red circle), ensuring high sensitivity while maintaining a stable baseline
against background seasonality. We also provided some examples of detected known events
(with maps) in those places from Li et al. (2022) (see Figs. R4-8).

Finally, to further address the reviewer's request using data from the current study, we have
added a new figure in the Supplementary Information (Supplementary Fig. 3) to show time series
plot of individual pixel could be used as a stable detector when observations are binned into
different view zenith angle intervals. In this specific case, the near nadir view (0-20 degrees)
showed the best range to identify this change (by the VZA-COLD algorithm).

References:

Olofsson, P., Foody, G. M., Herold, M., Stehman, S. V., Woodcock, C. E., & Wulder, M. A.
 (2014). Good practices for estimating area and assessing accuracy of land change. Remote
 sensing of Environment, 148, 42-57.
 Li, T., Zhu, Z., Wang, Z., Román, M. O., Kalb, V. L., & Zhao, Y. (2022). Continuous monitoring
 of nighttime light changes based on daily NASA's Black Marble product suite. Remote Sensing
 of Environment, 282, 113269.

 **Fig. R3.** Adapted from Fig. 3. in Li et al. (2022). The VZA stratified DNB time series (left) and
 the boxplot of DNB observations of different VZA intervals (right) for the corresponding pixel
 on the left. The red, green, and blue dots indicate the DNB observations within different VZA
 intervals. (c) Dense residential area with multi-story buildings, Nahyan, Egypt.

 **Fig. R4.** Adapted from Fig. 7. In Li et al., (2022). The VZA-COLD detection results for an
 urbanized suburban area in Melbourne, Australia. (a) The accumulated annual NTL change maps
 from 2013 to 2021 with the latest detected change year presented. (b) The day-of-year (DOY)
 NTL change maps over the region enlarged from the white rectangle from Fig. 7a. (c) The time

series plot of a selected pixel (red cross in Fig. 7b) and the corresponding VZA-COLD detection
 results, in which the red, green, blue, and grey colors indicate the different VZA intervals, the
 lines represent the estimated models, the small dots are the DNB observations, and the large
 magenta dot is the detected change. (d) The high-resolution Google Earth image in January 2015
 and March 2020, respectively. The red squares represent the location and size of the selected
 pixel in Fig. 7c. The grey background in Figs. 7a-b is the Esri ArcMap Dark Grey Canvas base
 map.

 **Fig. R5.** Adapted from Fig. 8. In Li et al., (2022). The VZA-COLD detection results of a selected
 pixel, and high-resolution images for a new airport built in Beijing, China. (a) The accumulated
 annual NTL change maps from 2013 to 2021 with the latest detected change year presented. (b)
 The day-of-year (DOY) NTL change map in 2019 over the region enlarged from the white
 rectangle from Fig. 8a. (c) The time series plot of a selected pixel (red cross in Fig. 8b) and the
 corresponding VZA-COLD detection results, in which the red, green, blue, and grey colors
 indicate the different VZA intervals, the lines represent the estimated models, the small dots are
 the DNB observations, and the large magenta dots are the detected changes. (d) The high-
 resolution Google Earth image in July 2015 and August 2020, respectively.

**Fig. R6.** Adapted from Fig. 11. In Li et al., (2022). The VZA-COLD detection results of a
 selected pixel, and high-resolution images for Milan, Italy. (a) The accumulated annual NTL
 change maps from 2013 to 2021 with the latest detected change year presented. (b) The day-of-
 798 year (DOY) NTL change map in 2014 over the region enlarged from the white rectangle from
 799 Fig. 11a. (c) The time series plot of a selected pixel (red cross in Fig. 11b) and the corresponding
 VZA-COLD detection results, in which the red, green, blue, and grey colors indicate the
 different VZA intervals, the lines represent the estimated models, the small dots are the DNB
 observations, and the large magenta dot is the detected changes. (d) The high-resolution Google
 Earth image in March 2013 and July 2015, respectively. The red squares represent the location
 and size of the selected pixel in Fig. 11c.

Fig. R7. Adapted from Fig. 12. In Li et al., (2022). The VZA-COLD detection results of a selected pixel, and high-resolution images for Puerto Rico. (a) The accumulated annual NTL change maps from 2013 to 2021 with the latest detected change year presented. (b) The day-of-day NTL change map in 2017 over the region enlarged from the white rectangle from Fig. 12a. (c) The time series plot of a selected pixel (red cross in Fig. 12b) and the corresponding VZA-COLD detection results, in which the red, green, blue, and grey colors indicate the different VZA intervals, the lines represent the estimated models, the small dots are the DNB observations, and the large magenta dots are the detected changes. (d) The high-resolution Google Earth image in July 2017 and October 2017. The red squares represent the location and size of the selected pixel in Fig. 12c.

**Fig. R8.** Adapted from Fig. 13. In Li et al., (2022). The VZA-COLD detection results of a
 selected pixel, and high-resolution images for Raqqa, Syria. (a) The accumulated annual NTL
 change maps from 2013 to 2021 with the latest detected change year presented. (b) The day-of-
 820 year (DOY) NTL change map in 2015 over the region enlarged from the white rectangle from
 821 Fig. 13a. (c) The time series plot of a selected pixel (red cross in Fig. 13b) and the corresponding
 VZA-COLD detection results, in which the red, green, blue, and grey colors indicate the
 different VZA intervals, the lines represent the estimated models, the small dots are the DNB
 observation, and the large magenta dots are the detected changes. (d) The high-resolution Google
 Earth image in January 2013 and July 2020, respectively.

**Referee #1 (Remarks to the Author):**

I would like to thank the authors for very thoroughly and successfully revising the text, based on
the review comments made.

**Response: We thank Referee #1 for the positive evaluation of the revised manuscript.**

**Referee #2 (Remarks to the Author):**

The authors have made a number of significant improvements. The revision of Manuscript #2025-
06-15916A moves the paper from very good to excellent. It now does justice to the outstanding
research it presents. I recommend it be published.

**Response: We thank Referee #2 for the positive assessment of the revision and for the**
**recommendation for publication.**